# Machine Unlearning of Federated Clusters

**Chao Pan**[*]**, Jin Sima**[*]**, Saurav Prakash**[*]**, Vishal Rana & Olgica Milenkovic**
Department of Electrical and Computer Engineering
University of Illinois Urbana-Champaign, USA
`{chaopan2,jsima,sauravp2,vishalr,milenkov}@illinois.edu`

## Abstract

Federated clustering (FC) is an unsupervised learning problem that arises in a number of practical applications, including personalized recommender and healthcare systems. With the adoption of recent laws ensuring the "right to be forgotten", the problem of machine unlearning for FC methods has become of significant importance. We introduce, for the first time, the problem of machine unlearning for FC, and propose an efficient unlearning mechanism for a customized secure FC framework. Our FC framework utilizes special initialization procedures that we show are well-suited for unlearning. To protect client data privacy, we develop the *secure compressed multiset aggregation (SCMA)* framework that addresses sparse secure federated learning (FL) problems encountered during clustering as well as more general problems. To simultaneously facilitate low communication complexity and secret sharing protocols, we integrate Reed-Solomon encoding with special evaluation points into our SCMA pipeline, and prove that the client communication cost is *logarithmic* in the vector dimension. Additionally, to demonstrate the benefits of our unlearning mechanism over complete retraining, we provide a theoretical analysis for the unlearning performance of our approach. Simulation results show that the new FC framework exhibits superior clustering performance compared to previously reported FC baselines when the cluster sizes are highly imbalanced. Compared to completely retraining $K$-means++ locally and globally for each removal request, our unlearning procedure offers an average speed-up of roughly 84x across seven datasets. Our implementation for the proposed method is available at `https://github.com/thupchnsky/mufc`.

## 1 Introduction

The availability of large volumes of user training data has contributed to the success of modern machine learning models. For example, most state-of-the-art computer vision models are trained on large-scale image datasets including Flickr (Thomee et al., 2016) and ImageNet (Deng et al., 2009). Organizations and repositories that collect and store user data must comply with privacy regulations, such as the recent European Union General Data Protection Regulation (GDPR), the California Consumer Privacy Act (CCPA), and the Canadian Consumer Privacy Protection Act (CPPA), all of which guarantee the right of users to remove their data from the datasets (*Right to be Forgotten*). Data removal requests frequently arise in practice, especially for sensitive datasets pertaining to medical records (numerous machine learning models in computational biology are trained using UK Biobank (Sudlow et al., 2015) which hosts a collection of genetic and medical records of roughly half a million patients (Ginart et al., 2019)). Removing user data from a dataset is insufficient to ensure sufficient privacy, since training data can often be reconstructed from trained models (Fredrikson et al., 2015; Veale et al., 2018). This motivates the study of *machine unlearning* (Cao & Yang, 2015) which aims to efficiently eliminate the influence of certain data points on a model. Naively, one can retrain the model from scratch to ensure complete removal, yet retraining comes at a high computational cost and is thus not practical when accommodating frequent removal requests. To avoid complete retraining, specialized approaches have to be developed for each unlearning application (Ginart et al., 2019; Guo et al., 2020; Bourtoule et al., 2021; Sekhari et al., 2021).

---

[*]Equal contribution.

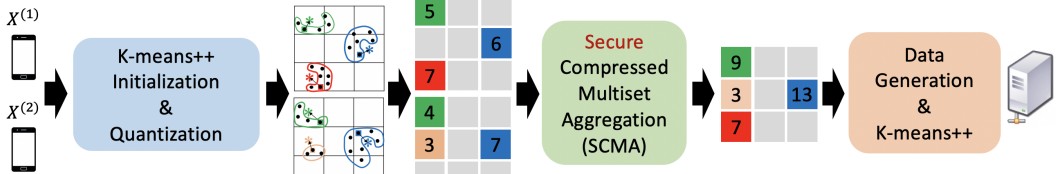

Figure 1: Overview of our proposed FC framework. $K$-means++ initialization and quantization are performed at each client in parallel. The SCMA procedure ensures that only the server knows the aggregated statistics of clients, without revealing who contributed the points in each individual cluster. The server generates points from the quantization bins with prescribed weights and performs full $K$-means++ clustering to infer the global model.

At the same time, federated learning (FL) has emerged as a promising approach to enable distributed training over a large number of users while protecting their privacy (McMahan et al., 2017; Chen et al., 2020; Kairouz et al., 2021; Wang et al., 2021; Bonawitz et al., 2021). The key idea of FL is to keep user data on their devices and train global models by aggregating local models in a communication-efficient and secure manner. Due to model inversion attacks (Zhu et al., 2019; Geiping et al., 2020), secure local model aggregation at the server is a critical consideration in FL, as it guarantees that the server cannot get specific information about client data based on their local models (Bonawitz et al., 2017; Bell et al., 2020; So et al., 2022; Chen et al., 2022). Since data privacy is the main goal in FL, it should be natural for a FL framework to allow for frequent data removal of a subset of client data in a cross-silo setting (e.g., when several patients request their data to be removed in the hospital database), or the entire local dataset for clients in a cross-device setting (e.g., when users request apps not to track their data on their phones). This leads to the largely unstudied problem termed *federated unlearning* (Liu et al., 2021; Wu et al., 2022; Wang et al., 2022). However, existing federated unlearning methods do not come with theoretical performance guarantees after model updates, and often, they are vulnerable to adversarial attacks.

Our contributions are summarized as follows. **1)** We introduce the problem of machine unlearning in FC, and design a new end-to-end system (Fig. 1) that performs highly efficient FC with privacy and low communication-cost guarantees, which also enables, when needed, simple and effective unlearning. **2)** As part of the FC scheme with unlearning features, we describe a novel one-shot FC algorithm that offers order-optimal approximation for the federated $K$-means clustering objective, and also outperforms the handful of existing related methods (Dennis et al., 2021; Ginart et al., 2019), especially for the case when the cluster sizes are highly imbalanced. **3)** For FC, we also describe a novel sparse compressed multiset aggregation (SCMA) scheme which ensures that the server only has access to the aggregated counts of points in individual clusters but has no information about the point distributions at individual clients. SCMA securely recovers the exact sum of the input sparse vectors with a communication complexity that is logarithmic in the vector dimension, outperforming existing sparse secure aggregation works (Beguier et al., 2020; Ergun et al., 2021), which have a linear complexity. **4)** We theoretically establish the unlearning complexity of our FC method and show that it is significantly lower than that of complete retraining. **5)** We compile a collection of datasets for benchmarking unlearning of federated clusters, including two new datasets containing methylation patterns in cancer genomes and gut microbiome information, which may be of significant importance to computational biologists and medical researchers that are frequently faced with unlearning requests. Experimental results reveal that our one-shot algorithm offers an average speed-up of roughly 84x compared to complete retraining across seven datasets.

## 2 RELATED WORKS

Due to space limitations, the complete discussion about related works is included in Appendix A.

**Federated clustering.** The goal of this learning task is to perform clustering using data that resides at different edge devices. Most of the handful of FC methods are centered around the idea of sending exact (Dennis et al., 2021) or quantized client (local) centroids (Ginart et al., 2019) directly to the server, which may not ensure desired levels of privacy as they leak the data statistics or cluster information of each individual client. To avoid sending exact centroids, Li et al. (2022) proposes sending distances between data points and centroids to the server without revealing the membership of data points to any of the parties involved, but their approach comes with large computational

and communication overhead. Our work introduces a novel communication-efficient secure FC framework, with a new privacy criterion that is intuitively appealing as it involves communicating obfuscated point counts of the clients to the server and frequently used in FL literature (Bonawitz et al., 2017).

**Machine unlearning.** Two types of unlearning requirements were proposed in previous works: exact unlearning (Cao & Yang, 2015; Ginart et al., 2019; Bourtoule et al., 2021; Chen et al., 2021) and approximate unlearning (Guo et al., 2020; Golatkar et al., 2020a;b; Sekhari et al., 2021; Fu et al., 2022; Chien et al., 2022). For exact unlearning, the unlearned model is required to perform identically as a completely retrained model. For approximate unlearning, the "differences" in behavior between the unlearned model and the completely retrained model should be appropriately bounded. A limited number of recent works also investigated data removal in the FL settings (Liu et al., 2021; Wu et al., 2022; Wang et al., 2022); however, most of them are empirical methods and do not come with theoretical guarantees for model performance after removal and/or for the unlearning efficiency. In contrast, our proposed FC framework not only enables efficient data removal in practice, but also provides theoretical guarantees for the unlearned model performance and for the expected time complexity of the unlearning procedure.

## 3 PRELIMINARIES

We start with a formal definition of the centralized $K$-means problem. Given a set of $n$ points in $\mathbb{R}^d$ $\mathcal{X}$ arranged into a matrix $X \in \mathbb{R}^{n \times d}$, and the number of clusters $K$, the $K$-means problem asks for finding a set of points $\mathbf{C} = \{c_1, ..., c_K\}, c_k \in \mathbb{R}^d, \forall k \in [K]$ that minimizes the objective

$$\phi_c(\mathcal{X}; \mathbf{C}) = \|X - C\|_F^2, \tag{1}$$

where $\|\cdot\|_F$ denotes the Frobenius norm of a matrix, $\|\cdot\|$ denotes the $\ell_2$ norm of a vector, and $C \in \mathbb{R}^{n \times d}$ records the closest centroid in $\mathbf{C}$ to each data point $x_i \in \mathcal{X}$ (i.e., $c_i = \arg\min_{c_j \in \mathbf{C}} \|x_i - c_j\|$). Without loss of generality, we make the assumption that the optimal solution is unique in order to facilitate simpler analysis and discussion, and denote the optimum by $\mathbf{C}^* = \{c_1^*, ..., c_K^*\}$. The set of centroids $\mathbf{C}^*$ induces an optimal partition $\bigcup_{k=1}^K \mathcal{C}_k^*$ over $\mathcal{X}$, where $\forall k \in [K], \mathcal{C}_k^* = \{x_i : \|x_i - c_k^*\| \leq \|x_i - c_j^*\| \ \forall i \in [n], j \in [K]\}$. We use $\phi_c^*(\mathcal{X})$ to denote the optimal value of the objective function for the centralized $K$-means problem. With a slight abuse of notation, we also use $\phi_c^*(\mathcal{C}_k^*)$ to denote the objective value contributed by the optimal cluster $\mathcal{C}_k^*$. A detailed description of a commonly used approach for solving the $K$-means problem, $K$-means++, is available in Appendix B.

In FC, the dataset $\mathcal{X}$ is no longer available at the centralized server. Instead, data is stored on $L$ edge devices (clients) and the goal of FC is to learn a global set of $K$ centroids $\mathbf{C}_s$ at the server based on the information sent by clients. For simplicity, we assume that there exists no identical data points across clients, and that the overall dataset $\mathcal{X}$ is the union of the datasets $\mathcal{X}^{(l)}$ arranged as $X^{(l)} \in \mathbb{R}^{n^{(l)} \times d}$ on device $l, \forall l \in [L]$. The server will receive the aggregated cluster statistics of all clients in a secure fashion, and generate the set $\mathbf{C}_s$. In this case, the federated $K$-means problem asks for finding $K$ global centroids $\mathbf{C}_s$ that minimize the objective

$$\phi_f(\mathcal{X}; \mathbf{C}_s) = \sum_{l=1}^L \|X^{(l)} - C_s^{(l)}\|_F^2, \tag{2}$$

where $C_s^{(l)} \in \mathbb{R}^{n^{(l)} \times d}$ records the centroids of the *induced global clusters* that data points $\{x_i^{(l)}\}_{i=1}^{n^{(l)}}$ on client $l$ belong to. Note that the definition of the assignment matrix $C$ for the centralized $K$-means is different from that obtained through federated $K$-means $C_s^{(l)}$: the $i$-th row of $C$ only depends on the location of $x_i$ while the row in $C_s^{(l)}$ corresponding to $x_i$ depends on the induced global clusters that $x_i$ belongs to (for a formal definition see 3.1). In Appendix L, we provide a simple example that further illustrates the difference between $C$ and $C_s^{(l)}$. Note that the notion of induced global clusters was also used in Dennis et al. (2021).

**Definition 3.1.** *Suppose that the local clusters at client $l$ are denoted by $\mathcal{C}_k^{(l)}, \forall k \in [K], l \in [L]$, and that the clusters at the server are denoted by $\mathcal{C}_k^s, \forall k \in [K]$. The global clustering equals $\mathcal{P}_k = \{x_i^{(l)} | x_i^{(l)} \in \mathcal{C}_j^{(l)}, c_j^{(l)} \in \mathcal{C}_k^s, \forall j \in [K], l \in [L]\}$, where $c_j^{(l)}$ is the centroid of $\mathcal{C}_j^{(l)}$ on client $l$. Note that $(\mathcal{P}_1, \ldots, \mathcal{P}_K)$ forms a partition of the entire dataset $\mathcal{X}$, and the representative centroid for $\mathcal{P}_k$ is defined as $c_{s,k} \in \mathbf{C}_s$.*

**Exact unlearning.** For clustering problems, the *exact unlearning* criterion may be formulated as follows. Let $\mathcal{X}$ be a given dataset and $\mathcal{A}$ a (randomized) clustering algorithm that trains on $\mathcal{X}$ and outputs a set of centroids $\mathbf{C} \in \mathcal{M}$, where $\mathcal{M}$ is the chosen space of models. Let $\mathcal{U}$ be an unlearning algorithm that is applied to $\mathcal{A}(\mathcal{X})$ to remove the effects of one data point $x \in \mathcal{X}$. Then $\mathcal{U}$ is an exact unlearning algorithm if $\forall \mathbf{C} \in \mathcal{M}, x \in \mathcal{X}, \mathbb{P}(\mathcal{U}(\mathcal{A}(\mathcal{X}), \mathcal{X}, x) = \mathbf{C}) = \mathbb{P}(\mathcal{A}(\mathcal{X} \backslash x) = \mathbf{C})$. To avoid confusion, in certain cases, this criterion is referred to as *probabilistic (model) equivalence.*

**Privacy-accuracy-efficiency trilemma.** How to trade-off data privacy, model performance, communication and computational efficiency is a long-standing problem in distributed learning (Acharya & Sun, 2019; Chen et al., 2020; Gandikota et al., 2021) that also carries over to FL and FC. Solutions that simultaneously address all these challenges in the latter context are still lacking. For example, Dennis et al. (2021) proposed a one-shot algorithm that takes model performance and communication efficiency into consideration by sending the *exact* centroids of each client to the server in a *nonanonymous* fashion. This approach may not be desirable under stringent privacy constraints as the server can gain information about individual client data. On the other hand, privacy considerations were addressed in Li et al. (2022) by performing $K$-means Lloyd's iterations anonymously via distribution of computations across different clients. Since the method relies on obfuscating pairwise distances for each client, it incurs computational overheads to hide the identity of contributing clients at the server and communication overheads due to interactive computations. None of the above methods is suitable for unlearning applications. To simultaneously enable unlearning and address the trilemma in the unlearning context, our privacy criterion involves transmitting *the number of client data points within local client clusters* in such a manner that the server cannot learn the data statistics of any specific client, but only the overall statistics of the union of client datasets. In this case, computations are limited and the clients on their end can perform efficient unlearning, unlike the case when presented with data point/centroid distances.

---

**Algorithm 1** Secure Federated Clustering

---

1: **input:** Dataset $\mathcal{X}$ distributed on $L$ clients $(\mathcal{X}^{(1)}, \ldots, \mathcal{X}^{(L)})$.

2: Run $K$-means++ initialization on each client $l$ in parallel, obtain the initial centroid sets $\mathbf{C}^{(l)}$, and record the corresponding cluster sizes $(|\mathcal{C}_1^{(l)}|, \ldots, |\mathcal{C}_K^{(l)}|), \ \forall l \in [L]$.

3: Perform uniform quantization of $\mathbf{C}^{(l)}$ on each dimension, and flatten the quantization bins into a vector $q^{(l)}, \forall l \in [L]$.

4: Set $q_j^{(l)} = \left| \mathcal{C}_k^{(l)} \right|$ with $j$ being the index of the quantization bin where $c_k^{(l)}$ lies in for $\forall k \in [K]$, and $c_k^{(l)}$ is the centroid of $\mathcal{C}_k^{(l)}$. Set $q_j^{(l)} = 0$ for all other indices.

5: Securely sum up $q^{(l)}$ at server by Algorithm 2, with the aggregated vector denoted as $q$.

6: For index $j \in \{t : q_t \neq 0\}$, sample $q_j$ points based on pre-defined distribution and denote their union as new dataset $\mathcal{X}_s$ at server.

7: Run full $K$-means++ clustering at server with $\mathcal{X}_s$ to obtain the centroid set $\mathbf{C}_s$ at server.

8: **return** Each client retains its own centroid set $\mathbf{C}^{(l)}$, server retains $\mathcal{X}_s, q$ and $\mathbf{C}_s$.

---

**Random and adversarial removal.** Most unlearning literature focuses on the case when all data points are equally likely to be removed, a setting known as *random removal*. However, adversarial data removal requests may arise when users are malicious in unlearning certain points that are critical for model training (i.e., boundary points in optimal clusters). We refer to such a removal request as *adversarial removal*. In Section 5, we provide theoretical analysis for both types of removal.

## 4 FEDERATED CLUSTERING WITH SECURE MODEL AGGREGATION

The block diagram of our FC (Alg. 1) is depicted in Fig. 1. It comprises five components: a client-side clustering, client local information processing, secure compressed aggregation, server data generation, and server-side clustering module. We explain next the role of each component of the system.

For client- and server-side clustering (line 2 and 7 of Alg. 1), we adopt $K$-means++ as it lends it itself to highly efficient unlearning, as explained in Section 5. Specifically, we only run the $K$-means++ initialization procedure at each client but full $K$-means++ clustering (initialization and Lloyd's algorithm) at the server.

Line 3 and 4 of Alg. 1 describe the procedure used to process the information of local client clusters. As shown in Fig. 1, we first quantize the local centroids to their closest centers of the quantization bins, and the spatial locations of quantization bins naturally form a tensor, in which we store the sizes of local clusters. A tensor is generated for each client $l$, and subsequently flattened to form a vector $q^{(l)}$. For simplicity, we use uniform quantization with step size $\gamma$ for each dimension (line 3 of

Alg. 1, with more details included in Appendix H). The parameter $\gamma > 0$ determines the number of quantization bins in each dimension. If the client data is not confined to the unit hypercube centered at the origin, we scale the data to meet this requirement. Then the number of quantization bins in each dimension equals $B = \gamma^{-1}$, while the total number of quantization bins for $d$ dimensions is $B^d = \gamma^{-d}$.

Line 5 of Alg. 1 describes how to aggregate information efficiently at the server without leaking individual client data statistics. This scheme is discussed in Section 4.1. Line 6 pertains to generating $q_j$ points for the $j$-th quantization bin based on its corresponding spatial location. The simplest idea is to choose the center of the quantization bin as the representative point and assign weight $q_j$ to it. Then, in line 7, we can use the weighted $K$-means++ algorithm at the server to further reduce the computational complexity.

A simplified version of Alg. 1 is discussed in Appendix I, for applications where the privacy criterion is not an imperative.

## 4.1 SCMA AT THE SERVER

**Algorithm 2** SCMA

1: **input:** $L$ different vectors $q^{(l)}$ of length $B^d$ to be securely aggregated, a finite field $\mathbb{F}_p$.
2: Each client $l \in [L]$ communicates $(S_1^{(l)}, \ldots, S_{2KL}^{(l)})$ to the server, where $S_i^{(l)} = (\sum_{j:q_j^{(l)} \neq 0} q_j^{(l)} \cdot j^{i-1} + z_i^{(l)}) \bmod p, i \in [2KL]$ and $z_i^{(l)}$ is a random key uniformly distributed over $\mathbb{F}_p$ and hidden from the server. The keys $\{z_i^{(l)}\}_{l \in [L], i \in [2KL]}$ are generated offline using standard secure model aggregation so that $(\sum_l z_i^{(l)}) \bmod p = 0$.
3: The server first computes the sum $S_i = (\sum_{l \in [L]} S_i^{(l)}) \bmod p$. Given $S_i$, the server computes the coefficients of the polynomial $g(x) = \prod_{j:q_j \neq 0} (1 - j \cdot x)$ using the Berlekamp-Massey algorithm (Berlekamp, 1968; Massey, 1969). Then, the server factorizes $g(x)$ over the field $\mathbb{F}_p$ to determine the roots $j^{-1}$, $q_j \neq 0$, using the polynomial factorizing algorithm (Kedlaya & Umans, 2011). Finally, the server solves a set of $2KL$ linear equations $S_i = \sum_{l \in [L]} S_i^{(l)} = \sum_{j:q_j \neq 0} q_j \cdot j^{i-1}$ for $i \in [2KL]$, by considering $q_j$ as unknowns and $j^{i-1}$ as known coefficients for $q_j \neq 0$.
4: **return** $q$ reconstructed at the server.

Once the vector representations $q^{(l)}$ of length $B^d$ for client $l$ are generated (line 4 of Alg. 1), we can use standard secure model aggregation methods (Bonawitz et al., 2017; Bell et al., 2020; So et al., 2022) to sum up all $q^{(l)}$ securely and obtain the aggregated results $q$ at the server. However, since the length of each vector $q^{(l)}$ is $B^d$, securely aggregating the whole vector would lead to an exponential communication complexity for each client. Moreover, each $q^{(l)}$ is a sparse vector since the number of client centroids is much smaller than the number of quantization bins (i.e., $K \ll B^d$). It is inefficient and unnecessary for each client to send out the entire $q^{(l)}$ with noisy masks for aggregation. This motivates us to first compress the vectors and then perform the secure aggregation, and we refer to this process as SCMA (Alg. 2), with one example illustrated in Fig. 2.

By observing that there can be at most $K$ nonzero entries in $q^{(l)}, \forall l \in [L]$ and at most $KL$ nonzero entries in $q$, we invoke the Reed-Solomon code construction (Reed & Solomon, 1960) for designing SCMA. Let $\mathbb{F}_p = \{0, 1, \ldots, p-1\}$ be a finite field of prime order $p \geq \max\{n, B^d\}$. We treat the indices of the quantization bins as distinct elements from the underlying finite field, and use them as evaluation points of the encoder polynomial. In addition, we treat a nonzero entry $q_j^{(l)}$ in vector $q^{(l)}$ as a substitution error at the $j$-th entry in a codeword. Then, we use our SCMA scheme shown in Alg. 2, where the messages that the clients send to server can be treated as syndromes in Reed-Solomon decoding. Note that in our scheme, the server does not know $q^{(l)}, l \in [L]$ beyond the fact that $\sum_{l \in [L]} q^{(l)} = q$, which fits into our privacy criterion. This follows because $z_i^{(l)}$ is uniformly distributed over $\mathbb{F}_p$ and independently chosen for different $l \in [L], i \in [2KL]$. For details, please refer to Appendix J.

## 4.2 PERFORMANCE ANALYSIS

We describe next the performance guarantees of Alg. 1 w.r.t. the objective defined in Eq. (2).

**Theorem 4.1.** *Suppose that we performed uniform quantization with step size $\gamma$ in Algorithm 1. Then we have* $\mathbb{E}\left(\phi_f(\mathcal{X}; \mathbf{C}_s)\right) < O(\log^2 K) \cdot \phi_c^*(\mathcal{X}) + O(nd\gamma^2 \log K)$.

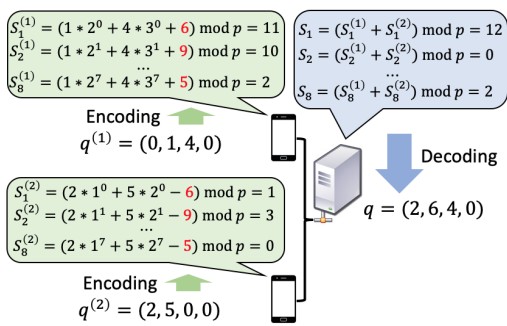

Figure 2: Example of the SCMA procedure for $K = 2, L = 2, B^d = 4, n = 12, p = 13$.

The performance guarantee in Theorem 4.1 pertains to two terms: the approximation of the optimal objective value and the quantization error (line 3 of Alg. 1). For the first term, the approximation factor $O(\log^2 K)$ is order-optimal for one-shot FC algorithms since one always needs to perform two rounds of clustering and each round will contribute a factor of $O(\log K)$. To make the second term a constant w.r.t. $n$, we can choose $\gamma = \Theta(1/\sqrt{n})$, which is a good choice in practice for the tested datasets as well. The above conclusions hold for any distribution of data across clients. Note that SCMA does not contribute to the distortion as it always returns the exact sum, while other methods for sparse secure aggregation based on sparsification (Han et al., 2020) may introduce errors and degrade the FC objective. See Appendix D for more details.

### 4.3 COMPLEXITY ANALYSIS

We derived a cohort of in-depth analysis pertaining to the computational and communication complexity for our proposed FC framework (Alg. 1). Due to space limitations, these results are summarized in Appendix C.

## 5 MACHINE UNLEARNING VIA SPECIALIZED SEEDING

We first describe an intuitive exact unlearning mechanism (Alg. 3) for $K$-means clustering in the centralized setting, which will be used later on as the unlearning procedure on the client-sides of the FC framework described in Section 5.3. The idea behind Alg. 3 is straightforward: one needs to rerun the $K$-means++ initialization, corresponding to retraining only if the current centroid set $\mathbf{C}$ contains at least one point requested for removal. This follows from two observations. First, since the centroids chosen through $K$-means++ initialization are true data points, the updated centroid set $\mathbf{C}'$ returned by Alg. 3 is guaranteed to contain no information about the data points that have been removed. Second, as we will explain in the next section, Alg. 3 also satisfies the exact unlearning criterion (defined in Section 3).

### 5.1 PERFORMANCE ANALYSIS

To verify that Alg. 3 is an exact unlearning method, we need to check that $\mathbf{C}'$ is probabilistically equivalent to the models generated by rerunning the $K$-means++ initialization process on $\mathcal{X}'$, the set of point remaining after removal. This is guaranteed by Lemma 5.1, and a formal proof is provided in Appendix E.

**Lemma 5.1.** *For any set of data points $\mathcal{X}$ and removal set $\mathcal{X}_R$, assuming that the remaining dataset is $\mathcal{X}' = \mathcal{X} \backslash \mathcal{X}_R$ and the centroid set returned by Algorithm 3 is $\mathbf{C}'$, we have*

$$\mathbb{P}(\mathcal{U}(\mathcal{A}(\mathcal{X}), \mathcal{X}, \mathcal{X}_R) = \mathbf{C}) = \mathbb{P}(\mathcal{A}(\mathcal{X}') = \mathbf{C}); \ \mathbb{E}(\phi_c(\mathcal{X}'; \mathbf{C}')) \le 8(\ln K + 2)\phi_c^*(\mathcal{X}'),$$

*where $\mathcal{A}$ represents Algorithm 1 and $\mathcal{U}$ represents the unlearning mechanism in Algorithm 3.*

### 5.2 COMPLEXITY ANALYSIS

We present next analytical results for the expected time complexity of removing a batch of $R$ data points simultaneously by our Alg. 3. For this, we consider both random and adversarial removal scenarios. While the analysis for random removal is fairly straightforward, the analysis for adversarial removal requests requires us to identify which removals force frequent retraining from scratch. In this regard, we state two assumptions concerning optimal cluster sizes and outliers, which will allow us to characterize the worst-case scenario removal setting.

**Assumption 5.2.** *Let $\epsilon_1 = \frac{n}{Ks_{\min}}$ be a constant denoting* cluster size imbalance, *where $s_{\min}$ equals the size of the smallest cluster in the optimal clustering; when $\epsilon_1 = 1$, all clusters are of size $\frac{n}{K}$.*

**Assumption 5.3.** *Assume that $\epsilon_2 \geq 1$ is a fixed constant. An outlier $x_i$ in $\mathcal{X}$ satisfies $\|x_i - c_j^*\| \leq \|x_i - c_k^*\|, \forall k \in [K]$ and $\|x_i - c_j^*\| > \sqrt{\epsilon_2 \phi_c^*(\mathcal{C}_j^*)/|\mathcal{C}_j^*|}$.*

---

**Algorithm 3** Unlearning via $K$-means++ Init.

---

1: **input:** Dataset $\mathcal{X}$, centroid set $\mathbf{C}$ obtained by $K$-means++ initialization on $\mathcal{X}$, removal request set $\mathcal{X}_R = \{x_{r_1}, \ldots, x_{r_R}\}$.

2: **if** $c_j \notin \mathcal{X}_R \ \forall c_j \in \mathbf{C}$ **then**
3:    $\mathbf{C}' \leftarrow \mathbf{C}$
4: **else**
5:    $i \leftarrow (\arg\min_j c_j \in \mathcal{X}_R) - 1$
6:    **if** $i = 0$ **then**
7:       $\mathbf{C}' \leftarrow \varnothing, \mathcal{X}' \leftarrow \mathcal{X} \backslash \mathcal{X}_R$.
8:    **else**
9:       $\mathbf{C}' \leftarrow \{c_1, \ldots, c_i\}, \mathcal{X}' \leftarrow \mathcal{X} \backslash \mathcal{X}_R$.
10:    **end if**
11:    **for** $j = i+1, \ldots, K$ **do**
12:       Sample $x$ from $\mathcal{X}'$ with prob $\frac{d^2(x, \mathbf{C}')}{\phi_c(\mathcal{X}'; \mathbf{C}')}$.
13:       $\mathbf{C}' \leftarrow \mathbf{C}' \cup \{x\}$.
14:    **end for**
15: **end if**
16: **return** $\mathbf{C}'$

---

Under Assumptions 5.2 and 5.3, we arrive at an estimate for the expected removal time presented in Theorem 5.4 below. Notably, the expected removal time does not depend on the data set size $n$.

**Theorem 5.4.** *Assume that the number of data points in $\mathcal{X}$ is $n$ and the probability of the data set containing at least one outlier is upper bounded by $O(1/n)$. Algorithm 3 supports removing $R$ points within one single request with expected time $\min\{O(RK^2d), O(nKd)\}$ for random removals, and expected time $\min\{O(RK^3\epsilon_1\epsilon_2 d), O(nKd)\}$ in expectation for adversarial removals. The complexity for complete retraining equals $O(nKd)$.*

**Remark.** *Due to the distance-based $K$-means++ initialization procedure, the existence of outliers in the dataset inevitably leads to higher retraining probability. This is the case since outliers are more likely to lie in the initial set of centroids. Hence, for analytical purposes, we assume in Theorem 5.4 that the probability of the data set containing at least one outlier is upper bounded by $O(1/n)$. This is not an overly restrictive assumption as there exist many different approaches for removing outliers before clustering Chawla & Gionis (2013); Gan & Ng (2017); Hautamäki et al. (2005), which effectively make the probability of outliers negligible.*

## 5.3 Unlearning Federated Clusters

We describe next the complete unlearning algorithm for the new FC framework which uses Alg. 3 for client-level clustering. In the FL setting, data resides on client storage devices, and thus the basic assumption of federated unlearning is that the removal requests will only appear at the client side, and the removal set will not be known to other unaffected clients and the server. We consider two types of removal requests in the FC setting: removing $R$ points from one client $l$ (cross-silo, single-client removal), and removing all data points from $R$ clients $l_1, \ldots, l_R$ (cross-device, multi-client removal). For the case where multiple clients want to unlearn only a part of their data, the approach is similar to that of single-client removal and can be handled via simple union bounds.

The unlearning procedure is depicted in Alg. 4. For single-client data removal, the algorithm will first perform unlearning at the client (say, client $l$) following Alg. 3. If the client's local clustering changes (i.e., client $l$ reruns the initialization), one will generate a new vector $q^{(l)}$ and send it to the server via SCMA. The server will rerun the clustering procedure with the new aggregated vector $q'$ and generate a new set of global centroids $\mathbf{C}'_s$. Note that other clients do not need to perform additional computations during this stage. For multi-client removals, we follow a similar strategy, except that no client needs to perform additional computations. Same as centralized unlearning described in Lemma 5.1, we can show that Alg. 4 is also an exact unlearning method.

**Removal time complexity.** For single-client removal, we know from Theorem 5.4 that the expected removal time complexity of client $l$ is $\min\{O(RK^2d), O(n^{(l)}Kd)\}$ and $\min\{O(RK^3\epsilon_1\epsilon_2 d), O(n^{(l)}Kd)\}$ for random and adversarial removals, respectively. $n^{(l)}$ denotes the number of data points on client $l$. Other clients do not require additional computations, since their centroids will not be affected by the removal requests. Meanwhile, the removal time complexity for the server is upper bounded by $O(K^2LTd)$, where $T$ is the maximum number of iterations of Lloyd's algorithm at the server before convergence. For multi-client removal, no client needs to perform additional computations, and the removal time complexity for the server equals $O((L-R)K^2Td)$.

# 6 EXPERIMENTAL RESULTS

**Algorithm 4** Unlearning of Federated Clusters

1: **input:** Dataset $\mathcal{X}$ distributed on $L$ clients $(\mathcal{X}^{(1)}, \ldots, \mathcal{X}^{(L)})$, $(\mathbf{C}^{(l)}, \mathcal{X}_s, q, \mathbf{C}_s)$ obtained by Algorithm 1 on $\mathcal{X}$, removal request set $\mathcal{X}_R^{(l)}$ for single-client removal or $\mathcal{L}_R$ for multi-client removal.
2: **if** single-client removal **then**
3:     Run Algorithm 3 on client $l$ and update $q^{(l)}$ if client $l$ has to perform retraining.
4: **else**
5:     $q^{(l)} \leftarrow \mathbf{0}$ on client $l$, $\forall l \in \mathcal{L}_R$.
6: **end if**
7: Securely sum up $q^{(l)}$ at server by Algorithm 2, with the aggregated vector denoted as $q'$.
8: **if** $q' = q$ **then**
9:     $\mathbf{C}'_s \leftarrow \mathbf{C}_s$.
10: **else**
11:     Generate $\mathcal{X}'_s$ with $q'$.
12:     Run full $K$-means++ at the server with $\mathcal{X}'_s$ to obtain $\mathbf{C}'_s$.
13: **end if**
14: **return** Client centroid sets $\mathbf{C}^{(l)\prime}$, server data $\mathcal{X}'_s, q'$ and centroids $\mathbf{C}'_s$.

To empirically characterize the trade-off between the efficiency of data removal and performance of our newly proposed FC method, we compare it with baseline methods on both synthetic and real datasets. Due to space limitations, more in-depth experiments and discussions are delegated to Appendix M.

**Datasets and baselines.** We use one synthetic dataset generated by a Gaussian Mixture Model (Gaussian) and six real datasets (Celltype, Covtype, FEMNIST, Postures, TMI, TCGA) in our experiments. We preprocess the datasets such that the data distribution is non-i.i.d. across different clients. The symbol $K'$ in Fig. 3 represents the maximum number of (true) clusters among clients, while $K$ represents the number of true clusters in the global dataset. A detailed description of the data statistics and the preprocessing procedure is available in Appendix M. Since there is currently no off-the-shelf algorithm designed for unlearning federated clusters, we adapt DC-Kmeans (DC-KM) from Ginart et al. (2019) to apply to our problem setting, and use complete retraining as the baseline comparison method. To evaluate FC performance on the complete dataset (before data removals), we also include the K-FED algorithm from Dennis et al. (2021) as the baseline method. In all plots, our Alg. 1 is referred to as MUFC. Note that in FL, clients are usually trained in parallel so that the estimated time complexity equals the sum of the longest processing time of a client and the processing time of the server.

**Clustering performance.** The clustering performance of all methods on the complete dataset is shown in the first row of Tab. 1. The loss ratio is defined as $\phi_f(\mathcal{X}; \mathbf{C}_s)/\phi_c^*(\mathcal{X})$[1], which is the metric used to evaluate the quality of the obtained clusters. For the seven datasets, MUFC offered the best performance on TMI and Celltype, datasets for which the numbers of data points in different clusters are highly imbalanced. This can be explained by pointing out an important difference between MUFC and K-FED/DC-KM: the quantized centroids sent by the clients may have non-unit weights, and MUFC is essentially performing weighted $K$-means++ at the server. In contrast, both K-FED and DC-KM assign equal unit weights to the client's centroids. Note that assigning weights to the client's centroids based on local clusterings not only enables a simple analysis of the scheme but also improves the empirical performance, especially for datasets with highly imbalanced cluster distributions. For all other datasets except Gaussian, MUFC obtained competitive clustering performance compared to K-FED/DC-KM. The main reason why DC-KM outperforms MUFC on Gaussian data is that all clusters are of the same size in this case. Also note that DC-KM runs full $K$-means++ clustering for each client while MUFC only performs initialization. Although running full $K$-means++ clustering at the client side can improve the empirical performance on certain datasets, it also greatly increases the computational complexity during training and the retraining probability during unlearning, which is shown in Fig. 3. Nevertheless, we also compare the performance of MUFC with K-FED/DC-KM when running full $K$-means++ clustering on clients for MUFC in Appendix M.

We also investigated the influence of $K'$ and $\gamma$ on the clustering performance. Fig. 3(a) shows that MUFC can obtain a lower loss ratio when $K' < K$, indicating that data is non-i.i.d. distributed across clients. Fig. 3(b) shows that the choice of $\gamma$ does not seem to have a strong influence on the clustering performance of Gaussian datasets, due to the fact that we use uniform sampling in Step 6 of Alg. 1 to generate the server dataset. Meanwhile, Fig. 3(c) shows that $\gamma$ can have a significant influence on the clustering performance of real-world datasets, which agrees with our analysis in Theorem 4.1.

---

[1]$\phi_c^*(X)$ is approximated by running $K$-means++ multiple times and selecting the smallest objective value.

Table 1: Clustering performance of different FC algorithms compared to centralized $K$-means++ clustering.

| | | TMI | Celltype | Gaussian | TCGA | Postures | FEMNIST | Covtype |
|---|---|---|---|---|---|---|---|---|
| Loss ratio | MUFC | $1.24 \pm 0.10$ | $1.14 \pm 0.03$ | $1.25 \pm 0.02$ | $1.18 \pm 0.05$ | $1.10 \pm 0.01$ | $1.20 \pm 0.00$ | $1.03 \pm 0.02$ |
| | K-FED | $1.84 \pm 0.07$ | $1.72 \pm 0.24$ | $1.25 \pm 0.01$ | $1.56 \pm 0.11$ | $1.13 \pm 0.01$ | $1.21 \pm 0.00$ | $1.60 \pm 0.01$ |
| | DC-KM | $1.54 \pm 0.13$ | $1.46 \pm 0.01$ | $1.02 \pm 0.00$ | $1.15 \pm 0.02$ | $1.03 \pm 0.00$ | $1.18 \pm 0.00$ | $1.03 \pm 0.02$ |
| Speed-up of MUFC (if no retraining is performed) | | 151x | 1535x | 2074x | 483x | 613x | 53x | 267x |

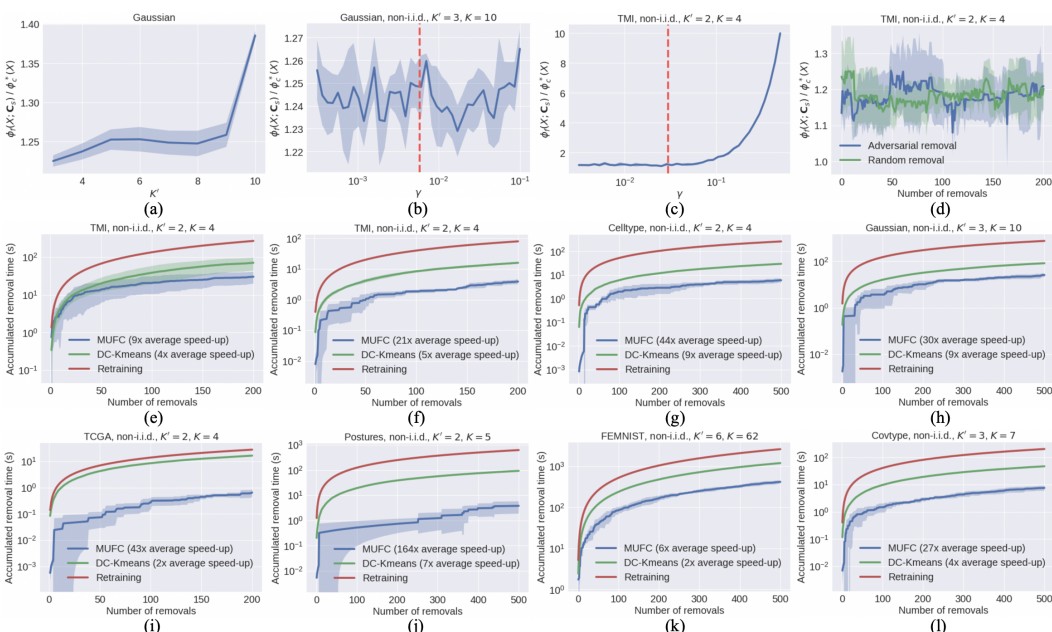

Figure 3: The shaded areas represent the standard deviation of results from different trails. (a) Influence of data heterogeneity on the clustering performance of MUFC: $K'$ represents the maximum number of (global) clusters covered by the data at the clients, while $K' = 10$ indicates that the data points are i.i.d. distributed across clients. (b)(c) Influence of the quantization step size $\gamma$ on the clustering performance of MUFC. The red vertical line indicates the default choice of $\gamma = 1/\sqrt{n}$, where $n$ is the total number of data points across clients. (d) The change in the loss ratio after each round of unlearning. (e) The accumulated removal time for adversarial removals. (f)-(l) The accumulated removal time for random removals.

The red vertical line in both figures indicates the default choice of $\gamma = 1/\sqrt{n}$, where $n$ stands for the number of total data points across clients.

**Unlearning performance.** Since K-FED does not support data removal, has high computational complexity, and its empirical clustering performance is worse than DC-KM (see Tab. 1), we only compare the unlearning performance of MUFC with that of DC-KM. For simplicity, we consider removing one data point from a uniformly at random chosen client $l$ at each round of unlearning. The second row of Tab. 1 records the speed-up ratio w.r.t. complete retraining for one round of MUFC unlearning (Alg. 4) when the removed point does not lie in the centroid set selected at client $l$. Fig. 3(e) shows the accumulated removal time on the TMI dataset for adversarial removals, which are simulated by removing the data points with the highest contribution to the current value of the objective function at each round, while Fig. 3(f)-(l) shows the accumulated removal time on different datasets for random removals. The results show that MUFC maintains high unlearning efficiency compared to all other baseline approaches, and offers an average speed-up ratio of $84$x when compared to complete retraining for random removals across seven datasets. We also report the change in the loss ratio of MUFC during unlearning in Fig. 3(d). The loss ratio remains nearly constant after each removal, indicating that our unlearning approach does not significantly degrade clustering performance. Similar conclusions hold for other tested datasets, as shown in Appendix M.

ETHICS STATEMENT

The seven datasets used in our simulations are all publicly available. Among these datasets, TCGA and TMI contain potentially sensitive biological data and are downloaded after logging into the database. We adhered to all regulations when handling this anonymized data and will only release the data processing pipeline and data that is unrestricted at TCGA and TMI. Datasets that do not contain sensitive information can be downloaded directly from their open-source repositories.

REPRODUCIBILITY STATEMENT

Our implementation is available at `https://github.com/thupchnsky/mufc`. Detailed instructions are included in the source code.

ACKNOWLEDGMENT

This work was funded by NSF grants 1816913 and 1956384. The authors thank Eli Chien for the helpful discussion.

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

## A  RELATED WORKS

**Federated clustering.** The idea of FC is to perform clustering using data that resides at different edge devices. It is closely related to clustered FL (Sattler et al., 2020), whose goal is to learn several global models simultaneously, based on the cluster structure of the dataset, as well as personalization according to the cluster assignments of client data in FL (Mansour et al., 2020). One difference between FC and distributed clustering (Guha et al., 2003; Ailon et al., 2009) is the assumption of data heterogeneity across different clients. Recent works (Ghosh et al., 2020; Dennis et al., 2021; Chung et al., 2022) exploit the non-i.i.d nature of client data to improve the performance of some learners. Another difference pertains to data privacy. Most previous methods were centered around the idea of sending exact (Dennis et al., 2021) or quantized client (local) centroids (Ginart et al., 2019) to the server, which may not be considered private as it leaks the data statistics or cluster information of all the clients. To avoid sending exact centroids, Li et al. (2022) proposes sending distances between data points and centroids to the server without revealing the membership of data points to any of the parties involved. Note that there is currently no formal definition of computational or information-theoretic secrecy/privacy for FC problems, making it hard to compare methods addressing different aspects of FL. Our method introduces a simple-to-unlearn clustering process and new privacy mechanism that is intuitively appealing as it involves communicating obfuscated point counts of the clients to the server.

**Sparse secure aggregation.** Sparse secure aggregation aims to securely aggregate local models in a communication-efficient fashion for the case that the local models are high-dimensional but sparse. Existing works on sparse secure aggregation (Beguier et al., 2020; Ergun et al., 2021) either have a communication complexity that is linear in the model dimension, or they can only generate an approximation of the aggregated model based on certain sparsification procedures (Han et al., 2020). In comparison, our SCMA scheme can securely recover the exact sum of the input sparse models with a communication complexity that is logarithmic in the model dimension.

**Private set union.** The private set union (Kissner & Song, 2005; Frikken, 2007; Seo et al., 2012) is a related but different problem compared to sparse secure aggregation. It requires multiple parties to communicate with each other to securely compute the union of their sets. In SCMA we aggregate multisets, which include the frequency of each element that is not considered in the private set union problem. In addition, our scheme includes only one round of communication from the clients to the server, while there is no server in the private set union problem but multi-round client to client communication is needed.

**Machine unlearning.** For *centralized* machine unlearning problems, two types of unlearning requirements were proposed in previous works: exact unlearning and approximate unlearning. For exact unlearning, the unlearned model is required to perform identically as a completely retrained model. To achieve this, Cao & Yang (2015) introduced distributed learners, Bourtoule et al. (2021) proposed sharding-based methods, Ginart et al. (2019) used quantization to eliminate the effect of removed data in clustering problems, and Chen et al. (2021) applied sharding-based methods to Graph Neural Networks. For approximate unlearning, the "differences" in behavior between the unlearned model and the completely retrained model should be appropriately bounded, similarly to what is done in the context of differential privacy. Following this latter direction, Guo et al. (2020) introduced the inverse Newton update for linear models, Sekhari et al. (2021) studied the generalization performance of approximately unlearned models, Fu et al. (2022) proposed an MCMC unlearning algorithm for sampling-based Bayesian inference, Golatkar et al. (2020a;b) designed model update mechanisms for deep neural networks based on Fisher Information and Neural Tangent Kernel, while Chien et al. (2022; 2023); Pan et al. (2023) extended the analysis to Graph Neural Networks. A limited number of recent works also investigated data removal in the FL settings: Liu et al. (2021) proposed to use fewer iterations during retraining for federated unlearning, Wu et al. (2022) introduced Knowledge Distillation into the unlearning procedure to eliminate the effect of data points requested for removal, and Wang et al. (2022) considered removing all data from one particular class via inspection of the internal influence of each channel in Convolutional Neural Networks. These federated unlearning methods are (mostly) empirical and do not come with theoretical guarantees for model performance after removal and/or for the unlearning efficiency. In contrast, our proposed FC framework not only enables efficient data removal in practice, but also provides theoretical guarantees for the unlearned model performance and for the expected time complexity of the unlearning procedure.

## B $K$-MEANS++ INITIALIZATION

The $K$-means problem is NP-hard even for $K = 2$, and when the points lie in a two-dimensional Euclidean space (Mahajan et al., 2012). Heuristic algorithms for solving the problem, including Lloyd's (Lloyd, 1982) and Hartigan's method (Hartigan & Wong, 1979), are not guaranteed to obtain the global optimal solution unless further assumptions are made on the point and cluster structures (Lee et al., 2017). Although obtaining the exact optimal solution for the $K$-means problem is difficult, there are many methods that can obtain quality approximations for the optimal centroids. For example, a randomized initialization algorithm ($K$-means++) was introduced in Vassilvitskii & Arthur (2006) and the expected objective value after initialization is a $(\log K)$-approximation to the optimal objective ($\mathbb{E}(\phi) \le (8 \ln K + 16)\phi^*$). $K$-means++ initialization works as follows: initially, the centroid set $\mathbf{C}$ is assumed to be empty. Then, a point is sampled uniformly at random from $X$ for the first centroid and added to $\mathbf{C}$. For the following $K - 1$ rounds, a point $x$ from $X$ is sampled with probability $d^2(x, \mathbf{C})/\phi_c(\mathcal{X}; \mathbf{C})$ for the new centroid and added to $\mathbf{C}$. Here, $d(x, \mathbf{C})$ denotes the minimum $\ell_2$ distance between $x$ and the centroids in $\mathbf{C}$ chosen so far. After the initialization step, we arrive at $K$ initial centroids in $\mathbf{C}$ used for running Lloyd's algorithm.

## C COMPLEXITY ANALYSIS OF ALGORITHM 1

**Computational complexity of client-side clustering.** Client-side clustering involves running $K$-means++ initialization procedure, which is of complexity $O(nKd)$.

**Computational complexity of server-side clustering.** Server-side clustering involves running $K$-means++ initialization procedure followed by Lloyd's algorithm with $T$ iterations, which is of complexity $O(K^2LTd)$.

**Computational complexity of SCMA at the client end.** The computation of $S_i^{(l)}$ on client $l$ requires at most $O(K \log i)$ multiplications over $\mathbb{F}_p$, $i \in [2KL]$. The total computational complexity equals $O(K^2L \log(KL))$ multiplication and addition operations over $\mathbb{F}_p$.

**Computational complexity of SCMA at the server end.** The computational complexity at the server is dominated by the complexity of the Berlekamp-Massey decoding algorithm (Berlekamp, 1968; Massey, 1969), factorizing the polynomial $g(x)$ over $\mathbb{F}_p$ (Kedlaya & Umans, 2011), and solving the linear equations $S_i = \sum_{l \in [L]} S_i^{(l)} = \sum_{j:q_j \ne 0} q_j \cdot j^{i-1}$ with known $j$, $q_j \ne 0$. The complexity of Berlekamp-Massey decoding over $\mathbb{F}_p$ is $O(K^2L^2)$. The complexity of factorizing a polynomial $g(x)$ over $\mathbb{F}_p$ using the algorithm in Kedlaya & Umans (2011) is $O((KL)^{1.5} \log p + KL \log^2 p)$ operations over $\mathbb{F}_p$. The complexity of solving for $S_i = \sum_{l \in [L]} S_i^{(l)}$ equals that of finding the inverse of a Vandermonde matrix, which takes $O(K^2L^2)$ operations over $\mathbb{F}_p$ (Eisinberg & Fedele, 2006). Hence, the total computational complexity at the server side is $\max\{O(K^2L^2), O((KL)^{1.5} \log p + KL \log^2 p)\}$ operations over $\mathbb{F}_p$.

**Communication complexity of SCMA at the client end.** Since each $S_i^{(l)}$ can be represented by $\lceil \log p \rceil$ bits, the information $\{S_i^{(l)}\}_{i \in [2KL]}$ sent by each client $l$ can be represented by $2KL\lceil \log p \rceil \le \max\{2KL \log n, 2KLd \log B\} + 1$ bits. Note that there are at most $\sum_{k \in [KL]} \binom{B^d}{k}\binom{n}{k-1}$ $q$-ary vectors of length $B^d$. Hence, the cost for communicating $q^{(l)}$ from the client to server $l$ is at least $\log\left(\sum_{k \in [KL]} \binom{B^d}{k}\binom{n}{k-1}\right) = \max\{O(KL \log n), O(KLd \log B)\}$ bits, which implies that our scheme is order-optimal w.r.t. the communication cost. Note that following standard practice in the area, we do not take into account the complexity of noise generation in secure model aggregation, as it can be done offline and independently of the Reed-Solomon encoding procedure.

## D PROOF OF THEOREM 4.1

*Proof.* We first consider the case where no quantization is performed (Algorithm 5). The performance guarantees for the federated objective value in this setting are provided in Lemma D.1.

**Lemma D.1.** *Suppose that the entire data set across clients is denoted by $\mathcal{X}$, and the set of server centroids returned by Algorithm 5 is $\mathbf{C}_s$. Then we have*

$$\mathbb{E}\left(\phi_f(\mathcal{X}; \mathbf{C}_s)\right) < O(\log^2 K) \cdot \phi_c^*(\mathcal{X}).$$

*Proof.* Let $\mathbf{C}^*$ denote the optimal set of centroids that minimize the objective (1) for the entire dataset $X \in \mathbb{R}^{n \times d}$, let $C^* \in \mathbb{R}^{n \times d}$ be the matrix that records the closest centroid in $\mathbf{C}^*$ to each data point, $\mathbf{C}_s$ the set of centroids returned by Alg. 1, and $C_s \in \mathbb{R}^{n \times d}$ the matrix that records the corresponding centroid in $\mathbf{C}_s$ for each data point based on the global clustering defined in Definition 3.1. Since we perform $K$-means++ initialization on each client dataset, for client $l$ it holds

$$\mathbb{E}\left(\|X^{(l)} - C^{(l)}\|_F^2\right) \le (8\ln K + 16)\|X^{(l)} - C_*^{(l)}\|_F^2, \quad \forall l \in [L]$$
$$\le (8\ln K + 16)\|X^{(l)} - C^{*,(l)}\|_F^2 \tag{3}$$

where $C^{(l)} \in \mathbb{R}^{n^{(l)} \times d}$ records the closest centroid in $\mathbf{C}^{(l)}$ to each data point $x_i$ in $\mathcal{X}^{(l)}$, $C_*^{(l)}$ is the optimal solution that can minimize the local $K$-means objective for client $l$, and $C^{*,(l)}$ denotes the row in $C^*$ that corresponds to client $l$. Summing up (3) over all clients gives

$$\mathbb{E}\left(\sum_{l=1}^L \|X^{(l)} - C^{(l)}\|_F^2\right) \le (8\ln K + 16)\sum_{l=1}^L \|X^{(l)} - C^{*,(l)}\|_F^2. \tag{4}$$

At the server side the client centroids are reorganized into a matrix $X_s \in \mathbb{R}^{n \times d}$. The weights of the client centroids are converted to replicates of rows in $X_s$. Since we perform full $K$-means++ clustering at the server, it follows that

$$\mathbb{E}\left(\|X_s - C_s\|_F^2\right) = \mathbb{E}\left(\sum_{l=1}^L \|C^{(l)} - C_s^{(l)}\|_F^2\right)$$
$$\overset{(a)}{\le} (8\ln K + 16)\sum_{l=1}^L \mathbb{E}\left(\|C^{(l)} - C_{s,*}^{(l)}\|_F^2\right)$$
$$\le (8\ln K + 16)\sum_{l=1}^L \mathbb{E}\left(\|C^{(l)} - C^{*,(l)}\|_F^2\right), \tag{5}$$

where $C_{s,*} \in \mathbb{R}^{n \times d}$ is the optimal solution that minimizes the $K$-means objective at the server. It is worth pointing out that $C_{s,*}$ is different from $C^*$, as they are optimal solutions for different optimization objectives. Note that we still keep the expectation on RHS for $(a)$. The randomness comes from the fact that $C^{(l)}$ is obtained by $K$-means++ initialization, which is a probabilistic procedure.

Combining (4) and (5) results in

$$\mathbb{E}\left(\phi_f(\mathcal{X}; \mathbf{C}_s)\right) = \mathbb{E}\left(\sum_{l=1}^L \|X^{(l)} - C_s^{(l)}\|_F^2\right)$$
$$\le 2 \cdot \mathbb{E}\left[\sum_{l=1}^L \left(\|X^{(l)} - C^{(l)}\|_F^2 + \|C^{(l)} - C_s^{(l)}\|_F^2\right)\right]$$
$$\le (16\ln K + 32)\sum_{l=1}^L \left[\|X^{(l)} - C^{*,(l)}\|_F^2 + \mathbb{E}\left(\|C^{(l)} - C^{*,(l)}\|_F^2\right)\right]. \tag{6}$$

For $\mathbb{E}\left(\|C^{(l)} - C^{*,(l)}\|_F^2\right)$, we have

$$\mathbb{E}\left(\|C^{(l)} - C^{*,(l)}\|_F^2\right) \le 2 \cdot \mathbb{E}\left(\|C^{(l)} - X^{(l)}\|_F^2 + \|X^{(l)} - C^{*,(l)}\|_F^2\right)$$
$$= 2 \cdot \|X^{(l)} - C^{*,(l)}\|_F^2 + 2 \cdot \mathbb{E}\left(\|C^{(l)} - X^{(l)}\|_F^2\right). \tag{7}$$

Replacing (7) into (6) shows that $\mathbb{E}\left(\phi_f(\mathcal{X}; \mathbf{C}_s)\right) < O(\log^2 K) \cdot \phi_c^*(X)$, which completes the proof.

If we are only concerned with the performance of non-outlier points over the entire dataset, we can upper bound the term $\mathbb{E}\left(\sum_{l=1}^{L}\|C^{(l)} - C^{*,(l)}\|_F^2\right)$ by

$$\mathbb{E}\left(\sum_{l=1}^{L}\|C^{(l)} - C^{*,(l)}\|_F^2\right) \leq \epsilon_2 \phi_c^*(\mathcal{X}). \tag{8}$$

Here, we used the fact that rows of $C^{(l)}$ are all real data points sampled by the $K$-means++ initialization procedure. For each data point $x_i$, it holds that $\|x_i - c_i^*\|^2|\mathcal{C}_i^*| \leq \epsilon_2\phi_c^*(\mathcal{C}_i^*)$, where $x_i \in \mathcal{C}_i^*$. In this case, we arrive at $\mathbb{E}\left(\phi_f(\mathcal{X}_t; \mathbf{C}_s)\right) < O(\epsilon_2 \log K) \cdot \phi_c^*(\mathcal{X}_t)$, where $\mathcal{X}_t$ corresponds to all non-outlier points. $\qquad\square$

**Remark.** *In Theorem 4 of Guha et al. (2003) the authors show that for the distributed $K$-median problem, if we use a $O(b)$-approximation algorithm (i.e., $\phi \leq O(b) \cdot \phi^*$) for the $K$-median problem with subdatasets on distributed machines, and use a $O(c)$-approximation algorithm for the $K$-median problem on the centralized machine, the overall distributed algorithm achieves effectively a $O(bc)$-approximation of the optimal solution to the centralized $K$-median problem. This is consistent with our observation that Alg. 5 can offer in expectation a $O(\log^2 K)$-approximation to the optimal solution of the centralized $K$-means problem, since $K$-means++ initialization achieves a $O(\log K)$-approximation on both the client and server side.*

*We also point out that in Dennis et al. (2021) the authors assume that the exact number of clusters from the global optimal clustering on client $l$ is known and equal to $K^{(l)}$, and propose the $K$-FED algorithm which performs well when $K' = \max_{l \in [L]} K^{(l)} \leq \sqrt{K}$. The difference between $K'$ and $K$ represents the data heterogeneity across different clients. With a slight modifications of the proof, we can also obtain $\mathbb{E}\left(\phi_f(\mathcal{X}; \mathbf{C}_s)\right) < O(\log K \cdot \log K') \cdot \phi_c^*(\mathcal{X})$, when $K^{(l)}$ is known for each client beforehand, and perform $K^{(l)}$-means++ on client $l$ instead of $K$-means++ in Alg. 1. For the extreme setting where each client safeguards data of one entire cluster (w.r.t. the global optimal clustering ($L = K, K' = 1$)), the performance guarantee for Alg. 1 becomes $\mathbb{E}\left(\phi_f(\mathcal{X}; \mathbf{C}_s)\right) < O(1) \cdot \phi_c^*(\mathcal{X})$, which is the same as seeding each optimal cluster by a data point sampled uniformly at random from that cluster. From Lemma 3.1 of Vassilvitskii & Arthur (2006) we see that we can indeed have $\mathbb{E}\left(\phi_f(\mathcal{X}; \mathbf{C}_s)\right) = 2\phi_c^*(\mathcal{X})$, where the approximation factor does not depend on $K$. This shows that data heterogeneity across different clients can benefit the entire FC framework introduced.*

Next we show the proof for Theorem 4.1. Following the same idea as the one used in the proof of Lemma D.1, we arrive at

$$\mathbb{E}\left(\phi_f(\mathcal{X}; \mathbf{C}_s)\right) \leq 3 \cdot \mathbb{E}\left[\sum_{l=1}^{L}\left(\|X^{(l)} - C^{(l)}\|_F^2 + \|C^{(l)} - \widehat{C}^{(l)}\|_F^2 + \|\widehat{C}^{(l)} - C_s^{(l)}\|_F^2\right)\right], \quad (9)$$

where $\widehat{C}^{(l)}$ is the quantized version of $C^{(l)}$. The first term can be upper bounded in the same way as in Lemma D.1. For the second term, the distortion introduced by quantizing one point is bounded by $\frac{\sqrt{d}\gamma}{2}$, if we choose the center of the quantization bin as the reconstruction point. Therefore,

$$\mathbb{E}\left(\sum_{l=1}^{L}\|C^{(l)} - \widehat{C}^{(l)}\|_F^2\right) \leq n\left(\frac{\sqrt{d}\gamma}{2}\right)^2 = \frac{nd\gamma^2}{4}. \tag{10}$$

The third term can be bounded as

$$\mathbb{E}\left(\sum_{l=1}^{L}\|\widehat{C}^{(l)} - C_s^{(l)}\|_F^2\right) \leq (8\ln K + 16)\sum_{l=1}^{L}\mathbb{E}\left(\|\widehat{C}^{(l)} - C^{*,(l)}\|_F^2\right)$$

$$\mathbb{E}\left(\|\widehat{C}^{(l)} - C^{*,(l)}\|_F^2\right) \leq 3 \cdot \mathbb{E}\left(\|\widehat{C}^{(l)} - C^{(l)}\|_F^2 + \|C^{(l)} - X^{(l)}\|_F^2 + \|X^{(l)} - C^{*,(l)}\|_F^2\right). \quad (11)$$

Replacing (10) and (11) into (9) leads to

$$\mathbb{E}\left(\phi_f(\mathcal{X}; \mathbf{C}_s)\right) < O(\log^2 K) \cdot \phi_c^*(\mathcal{X}) + O(nd\gamma^2 \log K),$$

which completes the proof. Similar as in Lemma D.1, we can have that for non-outlier points $\mathcal{X}_t$, $\mathbb{E}\left(\phi_f(\mathcal{X}_t; \mathbf{C}_s)\right) < O(\epsilon_2 \log K) \cdot \phi_c^*(\mathcal{X}_t) + O(nd\gamma^2 \log K)$. $\qquad\square$

## E   Proof of Lemma 5.1

*Proof.* Assume that the number of data points in $\mathcal{X}$ is $n$, the size of $\mathcal{X}_R$ is $R$, and the initial centroid set for $\mathcal{X}$ is $\mathbf{C}$. We use induction to prove that $\mathbf{C}'$ returned by Alg. 3 is probabilistically equivalent to rerunning the $K$-means++ initialization on $\mathcal{X}' = \mathcal{X}\backslash\mathcal{X}_R$.

The base case of induction amounts to investigating the removal process for $c_1$, the first point selected by $K$-means++. There are two possible scenarios: $c_1 \in \mathcal{X}_R$ and $c_1 \notin \mathcal{X}_R$. In the first case, we will rerun the initialization process over $\mathcal{X}'$, which is equivalent to retraining the model. In the second case, since we know $c_1 \notin \mathcal{X}_R$, the probability of choosing $c_1$ from $\mathcal{X}$ as the first centroid equals the conditional probability

$$\frac{1}{n-R} = \mathbb{P}(\text{choose } c_1 \text{ from } \mathcal{X} \text{ as the first centroid}|c_1 \notin \mathcal{X}_R)$$
$$= \mathbb{P}(\text{choose } c_1 \text{ from } \mathcal{X}' \text{ as the first centroid}).$$

Next suppose that $K > 1$, $i = (\arg\min_j c_j \in \mathcal{X}_R) - 1$. The centroids $\mathbf{C}'_{i-1} = \{c'_1 = c_1, \ldots, c'_{i-1} = c_{i-1}\}$ returned by Alg. 3 can be viewed probabilistically equivalent to the model obtained from rerunning the initialization process over $\mathcal{X}'$ for the first $i-1$ rounds. Then we have

$$\mathbb{P}(\text{choose } c_i \text{ from } \mathcal{X} \text{ as } i\text{-th centroid}|c_i \notin \mathcal{X}_R) = \frac{\mathbb{P}(\text{choose } c_i \text{ from } \mathcal{X} \text{ as } i\text{-th centroid} \cap c_i \notin \mathcal{X}_R)}{\mathbb{P}(c_i \notin \mathcal{X}_R)}$$
$$\overset{(a)}{=} \frac{\mathbb{P}(\text{choose } c_i \text{ from } \mathcal{X} \text{ as } i\text{-th centroid})}{\mathbb{P}(c_i \notin \mathcal{X}_R)}$$
$$= \frac{d^2(c_i, \mathbf{C}'_{i-1})/\phi_c(\mathcal{X}; \mathbf{C}'_{i-1})}{1 - \sum_{x \in \mathcal{X}_R} d^2(x, \mathbf{C}'_{i-1})/\phi_c(\mathcal{X}; \mathbf{C}'_{i-1})}$$
$$= \frac{d^2(c_i, \mathbf{C}'_{i-1})/\phi_c(\mathcal{X}; \mathbf{C}'_{i-1})}{\phi_c(\mathcal{X}'; \mathbf{C}'_{i-1})/\phi_c(\mathcal{X}; \mathbf{C}'_{i-1})}$$
$$= \frac{d^2(c_i, \mathbf{C}'_{i-1})}{\phi_c(\mathcal{X}'; \mathbf{C}'_{i-1})}$$
$$= \mathbb{P}(\text{choose } c_i \text{ from } \mathcal{X}' \text{ as } i\text{-th centroid}),$$

where $(a)$ holds based on the definition of $i$, indicating that the $i$-th centroid is not in $\mathcal{X}_R$. Therefore, the centroid $c'_i = c_i$ returned by Alg. 3 can be seen as if obtained from rerunning the initialization process over $\mathcal{X}'$ in the $i$-th round. Again based on the definition of $i$, it is clear that for $j > i$, $c'_j$ are the centroids chosen by the $K$-means++ procedure over $\mathcal{X}'$. This proves our claim that $\mathbf{C}'$ returned by Alg. 3 is probabilistic equivalent to the result obtained by rerunning the $K$-means++ initialization on $\mathcal{X}'$.

Theorem 1.1 of Vassilvitskii & Arthur (2006) then establishes that

$$\mathbb{E}(\phi_c(\mathcal{X}'; \mathbf{C}')) \le 8(\ln K + 2)\phi_c^*(\mathcal{X}'), \tag{12}$$

which completes the proof. $\qquad\square$

## F   Proof of Theorem 5.4

*Proof.* We first analyze the probability of rerunning $K$-means++ initialization based on Alg. 3. Assumptions 5.2 and 5.3 can be used to derive an expression for the probability of $x_i \in \mathbf{C}$ (where $x_i$ is the point that needs to be unlearned), which also equals the probability of retraining.

**Lemma F.1.** *Assume that the number of data points in $\mathcal{X}$ is $n$ and that the probability of the data set containing at least one outlier is upper bounded by $O(1/n)$. Let $\mathbf{C}$ be the centroid set obtained by running $K$-means++ on $\mathcal{X}$. For an arbitrary removal set $\mathcal{X}_R \subseteq \mathcal{X}$ of size $R$, we have*

$$\text{for random removals: } \mathbb{P}(\mathcal{X}_R \cap \mathbf{C} \ne \varnothing) < O(RK/n);$$
$$\text{for adversarial removals: } \mathbb{P}(\mathcal{X}_R \cap \mathbf{C} \ne \varnothing) < O(RK^2\epsilon_1\epsilon_2/n).$$

*Proof.* Since outliers can be arbitrarily far from all true cluster points based on definition, during initialization they may be sampled as centroids with very high probability. For simplicity of analysis, we thus assume that outliers are sampled as centroids with probability 1 if they exist in the dataset, meaning that we will always need to rerun the $K$-means++ initialization when outliers exist in the complete dataset before any removals.

For random removals, where the point requested for unlearning, $x_i$, is drawn uniformly at random from $\mathcal{X}$, it is clear that $\mathbb{P}(x_i \in \mathbf{C}) = \frac{K}{n}$, since $\mathbf{C}$ contains $K$ distinct data points in $\mathcal{X}$.

For adversarial removals, we need to analyze the probability of choosing $x_i$ as the $(k+1)$-th centroid, given that the first $k$ centroids have been determined and $x_i \notin \mathbf{C}_k = \{c_1, \dots, c_k\}$. For simplicity we first assume that there is no outlier in $\mathcal{X}$. Then we have

$$\mathbb{P}(\text{choose } x_i \text{ from } \mathcal{X} \text{ as the } (k+1)\text{-th centroid}|\mathbf{C}_k) = \frac{d^2(x_i, \mathbf{C}_k)}{\sum_{y \neq x_i} d^2(y, \mathbf{C}_k) + d^2(x_i, \mathbf{C}_k)} \quad (13)$$

For the denominator $\sum_{y \neq x_i} d^2(y, \mathbf{C}_k) + d^2(x_i, \mathbf{C}_k)$, the following three observations are in place

$$\sum_{y \neq x_i} d^2(y, \mathbf{C}_k) + d^2(x_i, \mathbf{C}_k) \geq \phi_c^*(\mathcal{X}) \geq \phi_c^*(\mathcal{C}_i^*), x_i \in \mathcal{C}_i^*$$

$$\sum_{y \neq x_i} d^2(y, \mathbf{C}_k) + d^2(x_i, \mathbf{C}_k) \geq \sum_{y \neq x_i} d^2(y, \mathbf{C}^*)$$

$$\sum_{y \neq x_i} d^2(y, \mathbf{C}_k) + d^2(x_i, \mathbf{C}_k) \geq \sum_{y \neq x_i} d^2(y, \mathbf{C}_k).$$

Therefore,

$$\sum_{y \neq x_i} d^2(y, \mathbf{C}_k) + d^2(x_i, \mathbf{C}_k) \geq \frac{\phi_c^*(\mathcal{C}_i^*)}{5} + \frac{2}{5}\left(\sum_{y \neq x_i} d^2(y, \mathbf{C}^*) + d^2(y, \mathbf{C}_k)\right)$$

$$\overset{(a)}{\geq} \frac{1}{5}\left(\phi_c^*(\mathcal{C}_i^*) + \sum_{y \neq x_i} \|c_y - c_y^*\|^2\right), \quad (14)$$

where $c_y, c_y^*$ are the closest centroid in $\mathbf{C}_k$ and $\mathbf{C}^*$ to $y$, respectively. Here, $(a)$ is a consequence of the fact that $\|a - b\|^2 = \|a - c + c - b\|^2 \leq 2(\|a - c\|^2 + \|b - c\|^2)$. Since $x_i$ is not an outlier for $\mathcal{C}_i^*$ based on our assumption, we have

$$\phi_c^*(\mathcal{C}_i^*) \geq \frac{|\mathcal{C}_i^*|}{\epsilon_2}\|x_i - c_i^*\|^2 \geq \frac{n}{K\epsilon_1\epsilon_2}\|x_i - c_i^*\|^2.$$

Consequently,

$$\phi_c^*(\mathcal{C}_i^*) + \sum_{y \neq x_i} \|c_y - c_y^*\|^2 \geq \frac{|\mathcal{C}_i^*|}{\epsilon_2}\|x_i - c_i^*\|^2 + \sum_{y \in \mathcal{C}_i^*} \|c_y - c_y^*\|^2$$

$$= \frac{|\mathcal{C}_i^*|}{\epsilon_2}\|x_i - c_i^*\|^2 + \sum_{y \in \mathcal{C}_i^*} \|c_y - c_i^*\|^2. \quad (15)$$

For $\forall y \in \mathcal{C}_i^*$, it hold $\|x_i - c_i^*\|^2 + \|c_y - c_i^*\|^2 \geq \frac{1}{2}\|x_i - c_y\|^2 \geq \frac{1}{2}d^2(x_i, \mathbf{C}_k)$. Thus, (15) can be lower bounded by

$$\frac{|\mathcal{C}_i^*|}{\epsilon_2}\|x_i - c_i^*\|^2 + \sum_{y \in \mathcal{C}_i^*} \|c_y - c_i^*\|^2 \geq \frac{|\mathcal{C}_i^*|}{2\epsilon_2}d^2(x_i, \mathbf{C}_k) \geq \frac{n}{2K\epsilon_1\epsilon_2}d^2(x_i, \mathbf{C}_k). \quad (16)$$

Combining (16) and (14) we obtain

$$\sum_{y \neq x_i} d^2(y, \mathbf{C}_k) + d^2(x_i, \mathbf{C}_k) \geq \frac{n}{10K\epsilon_1\epsilon_2}d^2(x_i, \mathbf{C}_k).$$

Using this expression in (13) results in

$$\mathbb{P}(\text{choose } x_i \text{ from } \mathcal{X} \text{ as the } (k+1)\text{-th centroid}|\mathbf{C}_k) \leq \frac{10K\epsilon_1\epsilon_2}{n}, \tag{17}$$

which holds for $\forall k \in [K]$. Thus, the probability $\mathbb{P}(x_i \in \mathbf{C})$ can be computed as

$$\mathbb{P}(x_i \in \mathbf{C}) = \sum_{k=0}^{K-1} \mathbb{P}(\text{choose } x_i \text{ from } \mathcal{X} \text{ as the } (k+1)\text{-th centroid}|\mathbf{C}_k)\mathbb{P}(\mathbf{C}_k)$$

$$\leq \sum_{k=0}^{K-1} \mathbb{P}(\text{choose } x_i \text{ from } \mathcal{X} \text{ as the } (k+1)\text{-th centroid}|\mathbf{C}_k)$$

$$\leq \frac{1}{n} + \frac{10K(K-1)\epsilon_1\epsilon_2}{n} < O\left(\frac{K^2\epsilon_1\epsilon_2}{n}\right). \tag{18}$$

Here, we assumed that $\mathbf{C}_0 = \varnothing$.

For the case where outliers are present in the dataset, we have

$$\mathbb{P}(x_i \in \mathbf{C}) = \mathbb{P}(x_i \in \mathbf{C}|x_i \text{ is outlier})\mathbb{P}(x_i \text{ is outlier}) + \mathbb{P}(x_i \in \mathbf{C}|x_i \text{ is not outlier})\mathbb{P}(x_i \text{ is not outlier})$$

$$\leq 1 \cdot O\left(\frac{1}{n}\right) + O\left(\frac{K^2\epsilon_1\epsilon_2}{n}\right) \cdot 1 < O\left(\frac{K^2\epsilon_1\epsilon_2}{n}\right),$$

which completes the proof for the adversarial removal scenario. Finally, by union bound we can have that for the removal set $\mathcal{X}_R$ of size $R$,

$$\text{random removals: } \mathbb{P}(\mathcal{X}_R \cap \mathbf{C} \neq \varnothing) < O\left(\frac{RK}{n}\right);$$

$$\text{adversarial removals: } \mathbb{P}(\mathcal{X}_R \cap \mathbf{C} \neq \varnothing) < O\left(\frac{RK^2\epsilon_1\epsilon_2}{n}\right).$$

Also, the probability naturally satisfies that

$$\mathbb{P}(\mathcal{X}_R \cap \mathbf{C} \neq \varnothing) \leq 1.$$

$\square$

Next we show the proof for Theorem 5.4. The expected removal time for random removals can be upper bounded by

$$\mathbb{E}(\text{Removal time}) = \mathbb{E}(\text{Removal time}|\text{new initialization needed})\mathbb{P}(\text{new initialization needed}) +$$

$$\mathbb{E}(\text{Removal time}|\text{new initialization not needed})\mathbb{P}(\text{new initialization not needed})$$

$$\leq O(nKd + RK) \cdot O\left(\frac{RK}{n}\right) + O(RK) \cdot 1$$

$$< O(RK^2 d).$$

Following a similar argument, we can also show that the expected removal time for adversarial removals can be upper bounded by $O(RK^3\epsilon_1\epsilon_2 d)$. And based on our Algorithm 3, the unlearning complexity for both types of removal requests would be always upper bounded by the retraining complexity $O(nKd)$ as well, which completes the proof. $\square$

## G  COMPARISON BETWEEN ALGORITHM 3 AND QUANTIZED $K$-MEANS

In Ginart et al. (2019), quantized $K$-means were proposed to solve a similar problem of machine unlearning in the centralized setting. However, that approach substantially differs from Alg. 3. First, the intuition behind quantized $K$-means is that the centroids are computed by taking an average, and the effect of a small number of points is negligible when there are enough terms left in the clusters after removal. Therefore, if we quantize all centroids after each Lloyd's iteration, the quantized centroids will not change with high probability when we remove a small number of points from the

dataset. Meanwhile, the intuition behind Alg. 3 is as described in Lemma F.1. Second, the expected removal time complexity for quantized $K$-means equals $O\left(R^2 K^3 T^2 d^{2.5}/\epsilon\right)$, which is high since one needs to check if all quantized centroids remain unchanged after removal at each iteration, where $T$ denotes the maximum number of Lloyd's iteration before convergence and $\epsilon$ is some intrinsic parameter. In contrast, Alg. 3 only needs $O(RK^3 \epsilon_1 \epsilon_2 d)$ even for adversarial removals. Also note that the described quantized $K$-means algorithm does not come with performance guarantees on removal time complexity unless it is randomly initialized.

## H QUANTIZATION

For uniform quantization, we set $\hat{y} = \gamma \cdot a(y)$, where $a(y) = \arg\min_{j\in\mathbb{Z}} |y - \gamma j|, y \in \mathbb{R}^2$. The parameter $\gamma > 0$ determines the number of quantization bins in each dimension. Suppose all client data lie in the unit hypercube centered at the origin, and that if needed, pre-processing is performed to meet this requirement. Then the number of quantization bins in each dimension equals $B = \gamma^{-1}$, while the total number of quantization bins for $d$ dimensions is $B^d = \gamma^{-d}$.

In Section 4, we remarked that one can generate $q_j$ points by choosing the center of the quantization bin as the representative point and endow it with a weight equal to $q_j$. Then, in line 7, we can use the weighted $K$-means++ algorithm at the server to further reduce the computational complexity, since the effective problem size at the server reduces from $n$ to $KL$. However, in practice we find that when the computational power of the server is not the bottleneck in the FL system, generating data points uniformly at random within the quantization bins can often lead to improved clustering performance. Thus, this is the default approach for our subsequent numerical simulations.

## I SIMPLIFIED FEDERATED $K$-MEANS CLUSTERING

When privacy criterion like the one stated in Section 3 is not enforced, and as done in the framework of Dennis et al. (2021), one can skip line 3-6 in Alg. 1 and send the centroid set $\mathbf{C}^{(l)}$ obtained by client $l$ along with the cluster sizes $(|\mathcal{C}_1^{(l)}|, \ldots, |\mathcal{C}_K^{(l)}|)$ directly to the server. Then, one can run the weighted $K$-means++ algorithm at the server on the aggregated centroid set to obtain $\mathbf{C}_s$. The pseudocode for this simplified case is shown in Alg. 5. It follows a similar idea as the divide-and-conquer schemes of Guha et al. (2003); Ailon et al. (2009), developed for distributed clustering.

---

**Algorithm 5** Simplified Federated $K$-means Clustering

1: **input:** Dataset $\mathcal{X}$ distributed on $L$ clients $(\mathcal{X}^{(1)}, \ldots, \mathcal{X}^{(L)})$.
2: Run $K$-means++ initialization on each client $l$ in parallel, obtain the initial centroid sets $\mathbf{C}^{(l)}$, and record the corresponding cluster sizes $\left(|\mathcal{C}_1^{(l)}|, \ldots, |\mathcal{C}_K^{(l)}|\right)$, $\forall l \in [L]$.
3: Send $\left(c_1^{(l)}, \ldots, c_K^{(l)}\right)$ along with the corresponding cluster sizes $\left(|\mathcal{C}_1^{(l)}|, \ldots, |\mathcal{C}_K^{(l)}|\right)$ to the server, $\forall l \in [L]$.
4: Concatenate $\left(c_1^{(l)}, \ldots, c_K^{(l)}\right)$ as rows of $X_s$ and set $\left(|\mathcal{C}_1^{(l)}|, \ldots, |\mathcal{C}_K^{(l)}|\right)$ as the weights for the corresponding rows, $\forall l \in [L]$.
5: Run full weighted $K$-means++ clustering at server with $X_s$ to obtain the centroid set at server $\mathbf{C}_s$.
6: **return** Each client retains their own centroid set $\mathbf{C}^{(l)}$ while the server retains $X_s$ and $\mathbf{C}_s$.

---

In line 5 of Alg. 5, weighted $K$-means++ would assign weights to data points when computing the sampling probability during the initialization procedure and when computing the average of clusters during the Lloyd's iterations. Since the weights we are considering here are always positive integers, a weighted data point can also be viewed as there exist identical data points in the dataset with multiplicity equals to the weight.

---

[2] We can also add random shifts during quantization as proposed in Ginart et al. (2019) to make the data appear more uniformly distributed within the quantization bins.

## J  THE UNIQUENESS OF THE VECTOR $q$ GIVEN $\{S_i\}_{i \in [2KL]}$

To demonstrate that the messages generated by Alg. 2 can be uniquely decoded, we prove that there exists a unique $q$ that produces the aggregated values $\{S_i\}_{i \in [2KL]}$ at the server. The proof is by contradiction. Assume that there exist two different vectors $q$ and $q'$ that result in the same $\{S_i\}_{i \in [2KL]}$. In this case, we have the following set of linear equations $\sum_{j:q_j \neq 0} q_j \cdot j^{i-1} - \sum_{j:q'_j \neq 0} q'_j \cdot j^{i-1} = 0$, $i \in [2KL]$. Given that $\{q_j : q_j \neq 0\}$ and $\{q'_j : q'_j \neq 0\}$ represent at most $2KL$ unknowns and $j^{i-1}$ coefficients, the linear equations can be described using a square Vandermonde matrix for the coefficients, with the columns of the generated by the indices of the nonzero entries in $q$. This leads to a contradiction since a square Vandermonde matrix with different column generators is invertible, which we show below. Hence, the aggregated values $\{S_i\}$ must be different for different $q$. Similarly, the sums $\sum_{j:q_j^{(l)} \neq 0} q_j^{(l)} \cdot j^{i-1}$ are distinct for different choices of vectors $q^{(l)}$, $i \in [2KL]$, $l \in [L]$.

If two vectors $q$ and $q'$ result in the same $\{S_i\}_{i \in [2KL]}$, then $\sum_{j:q_j \neq 0} q_j \cdot j^{i-1} - \sum_{j:q'_j \neq 0} q'_j \cdot j^{i-1} = 0$, for all $i \in [2KL]$. Let $\{i_1, \ldots, i_u\} = (\{j : q_j \neq 0\} \cup \{j : q'_j = 0\})$ be the set of integers such that at least one of $q_{i_m}$ and $q'_{i_m}$ is nonzero for $m \in [u]$. Note that $u \leq 2KL$. Rewrite this equation as

$$\begin{bmatrix} 1 & \cdots & 1 \\ i_1 & \cdots & i_u \\ \vdots & \vdots & \vdots \\ i_1^{2KL-1} & \cdots & i_u^{2KL-1} \end{bmatrix} \begin{bmatrix} q_{i_1} - q'_{i_1} \\ \vdots \\ q_{i_u} - q'_{i_u} \end{bmatrix} = \mathbf{0}. \tag{19}$$

Since $u \leq 2KL$, we take the first $u$ equations in (19) and rewrite them as

$$Bv = \mathbf{0},$$

where

$$B = \begin{bmatrix} 1 & \cdots & 1 \\ i_1 & \cdots & i_u \\ \vdots & \vdots & \vdots \\ i_1^{2KL-1} & \cdots & i_u^{2KL-1} \end{bmatrix}$$

is a square Vandermonde matrix and

$$v = \begin{bmatrix} q_{i_1} - q'_{i_1} \\ \vdots \\ q_{i_u} - q'_{i_u} \end{bmatrix}$$

is a nonzero vector since $q \neq q'$. It is known that the determinant of a square Vandermonde matrix $B$ is given by $\prod_{m_1 < m_2, m_1, m_2 \in [u]} (i_{m_2} - i_{m_1})$, which in our case is nonzero since all the $i_1, \ldots, i_u$ are different. Therefore, $B$ is invertible and does not admit a non-zero solution, which contradicts the equation $Bv = \mathbf{0}$.

## K  A DETERMINISTIC LOW-COMPLEXITY ALGORITHM FOR SCMA AT THE SERVER

In the SCMA scheme we described in Alg. 1, the goal of the server is to reconstruct the vector $q$, given values $S_i = \sum_{j:q_j \neq 0} q_j \cdot j^{i-1} \mod p$ for $i \in [2KL]$. To this end, we first use the Berlekamp-Massey algorithm to compute the polynomial $g(x) = \prod_{j:q_j \neq 0} (1 - j \cdot x)$. Then, we factorize $g(x)$ over the finite field $\mathbb{F}_p$ using the algorithm described in Kedlaya & Umans (2011). The complexity $O((KL)^{1.5} \log p + KL \log^2 p)$ referred to in Section 4.3 corresponds to the average complexity (finding a deterministic algorithm that factorizes a polynomial over finite fields with $poly(\log p)$ worst-case complexity is an open problem). The complexity $\max\{O(K^2 L^2), O((KL)^{1.5} \log p + KL \log^2 p)\}$ referred to in Appendix C for the SCMA scheme represents an average complexity.

We show next that the SCMA scheme has small worst-case complexity under a deterministic decoding algorithm at the server as well. To this end, we replace the integer $p$ in Alg. 2 with a large number $p' \geq \max\{KLB^{2dKL}, n\} + 1$ such that $p'$ is larger than the largest possible $S_i$ and there is no overflow when applying the modulo $p'$ operation on $S_i$. It is known (Bertrand's postulate) that there exists a prime number between any integer $n > 3$ and $2n - 2$, and hence there must be a prime number lower-bounded by $\max\{KLB^{2dKL}, n\} + 1$ and twice the lower bound $2(\max\{KLB^{2dKL}, n\} + 1)$. However, since searching for a prime number of this size can be computationally intractable, we remove the requirement that $p'$ is prime. Correspondingly, $\mathbb{F}_{p'}$ is not necessarily a finite field. Then, instead of sending $S_i^{(l)} = (\sum_{j:q_j^{(l)} \neq 0} q_j^{(l)} \cdot j^{i-1} + z_i^{(l)}) \bmod p$, client $l$, $l \in [L]$, will send $S_i^{(l)} = (\sum_{j:q_j^{(l)} \neq 0} q_j^{(l)} \cdot j^{i-1} + z_i^{(l)}) \bmod p'$ to the server, $i \in [2KL]$, where random keys $z_i^{(l)}$ are independently and uniformly distributed over $\{0, \ldots, p' - 1\}$ and hidden from the server. After obtaining $S_i$, $i \in [2KL]$, the server can continue performing operations over the field of reals since there is no overflow in computing $S_i \bmod p'$. We note that though $p'$ is exponentially large, the computation of $S_i^{(l)}$ and $S_i$, $l \in [L]$ and $i \in [2KL]$ is still manageable, and achieved by computing and storing $S_i^{(l)}$ and $S_i$ using $O(KL)$ floating point numbers, instead of computing and storing $S_i^{(l)}$ in a single floating point number. Note that $j^i$ can be computed using $O(i)$ floating point numbers with complexity almost linear in $i$ (i.e., $O(i \log^c i)$ for some constant $c$).

We now present a low complexity secure aggregation algorithm at the server. After reconstructing $S_i$, we have $S_i = \sum_{j:q_j \neq 0} q_j \cdot j^{i-1}$. The server switches to computations over the real field. First, it uses the Berlekamp-Massey algorithm to find the polynomial $g(x) = \prod_{j:q_j \neq 0} (1 - j \cdot x)$ (the algorithm was originally proposed for decoding of BCH codes over finite fields, but it applies to arbitrary fields). Let $m$ be the degree of $g(x)$. Then $h(x) = x^m g(1/x) = \prod_{j:q_j \neq 0} (x - j)$. The goal is to factorize $h(x)$ over the field of reals, where the roots are known to be integers in $[B^d]$ and the multiplicity of each root is one.

If the degree of $h(x)$ is odd, then $h(0) < 0$ and $h(B^d) > 0$. Then we can use bisection search to find a root of $h(x)$, which requires $O(\log B^d)$ polynomial evaluations of $h(x)$, and thus $O(MK \log B^d)$ multiplication and addition operations of integers of size at most $\log p'$. After finding one root $j$, we can divide $h(x)$ by $x - j$ and start the next root-finding iteration.

If the degree of $h(x)$ is even, then the degree of $h'(x)$ is odd, and the roots of $h'(x)$ are different and confined to $[B^d]$. We use bisection search to find a root $j'$ of $h'(x)$. If $h(j') < 0$, then we use bisection search on $[0, j'] = \{0, 1, \ldots, j'\}$ to find a root of $h(x)$ and start a new iteration as described above when the degree of $h(x)$ is odd. If $h(j') > 0$, then $h'(j' - 1) > 0$ and $h'(0) < 0$. We use bisection search to find another root of $h'(x)$ in $[j' - 1]$. Note that for every two roots $j_1'$ and $j_2'$ ($j_1' < j_2'$) of $h'(x)$ satisfying $h(j_1') > 0$ and $h(j_2') > 0$ we can always find another root $j_3'$ of $h'(x)$ in $[j_1' + 1, j_2' - 1]$. We keep iterating the search for every two such roots $j_1', j_2'$ until we find a list of roots $r_1, \ldots, r_{2R+1}$ of $h'(x)$ such that $h(r_i) < 0$ for odd $i$ in $[2R + 1]$ and $h(r_i) > 0$ for even $i \in [2R + 1]$. Then we can run bisection search on the sets $[0, r_1], [r_1, r_2], \ldots, [r_{2R}, r_{2R+1}], [r_{2R+1}, B^d]$, to find $2R + 2$ roots of $h(x)$. Note that during the iteration we need $2R + 1$ bisection search iterations to find the roots $r_1, \ldots, r_{2R+1}$ for $h'(x)$ and $2R + 2$ bisection search iterations to find $2R + 2$ roots for $h(x)$.

The total computations complexity is hence at most $O(MK \log B^d)$ evaluations of polynomials with degree at most $O(MK)$ and at most $O(MK)$ polynomial divisions, which requires at most $O((MK)^2 \log B^d)$ multiplications and additions for integers of size at most $\log p'$. This results in an overall complexity of $O((MK)^3 d^2 \log^c(MK) \log B)$, for some constant $c < 2$.

## L    DIFFERENCE BETWEEN THE ASSIGNMENT MATRICES $C$ AND $C_s$

One example that explains the difference between these two assignment matrices is as follows. Suppose the global data sets and centroid sets are the same for the centralized and FC settings, i.e.,

$$X = \begin{bmatrix} X^{(1)} \\ \cdots \\ X^{(L)} \end{bmatrix}, \ \mathbf{C} = \mathbf{C}_s = \{c_1, \ldots, c_K\}.$$

Suppose that for $x_1$, which is the first row of $X$, we have

$$d(x_1, c_1) < d(x_1, c_j), \ \forall j \in [K], j \neq 1.$$

Then, the first row of $C$ equals $c_1$. However, if $x_1$ resides on the memory of client $l$ and belongs to the local cluster $\mathcal{C}_i^{(l)}$, and the recorded local centroid $c_i^{(l)}$ satisfies

$$d\left(c_i^{(l)}, c_2\right) < d\left(c_i^{(l)}, c_j\right), \ \forall j \in [K], j \neq 2,$$

then the first row of $C_s$ is $c_2$, even if $d(x_1, c_1) < d(x_1, c_2)$. Here $C_s$ is the row concatenation of the matrices $C_s^{(l)}$ on client $l$. This example shows that the assignment matrices $C$ and $C_s$ are different, which also implies that $\phi_f$ and $\phi_c$ are different.

# M    EXPERIMENTAL SETUP AND ADDITIONAL RESULTS

## M.1    DATASETS

In what follows, we describe the datasets used in our numerical experiments. Note that we preprocessed all datasets such that the absolute value of each element in the data matrix is smaller than 1. Each dataset has an intrinsic parameter $K$ for the number of optimal clusters, and these are used in the centralized $K$-means++ algorithm to compute the approximation of the optimal objective value. We use $\phi_c^*(X)$ in subsequent derivation to denote the objective value returned by the $K$-means++ algorithm. Besides $K$, we set an additional parameter $K' \sim \sqrt{K}$ for each client data so that the number of true clusters at the client level is not larger than $K'$. This non-i.i.d. data distribution across clients is discussed in Dennis et al. (2021). For small datasets (e.g., TCGA, TMI), we consider the number of clients $L$ as 10, and set $L = 100$ for all other datasets.

**Celltype** [$n = 12009, d = 10, K = 4$] (Han et al., 2018; Gardner et al., 2014b) comprises single cell RNA sequences belonging to a mixture of four cell types: fibroblasts, microglial cells, endothelial cells and mesenchymal stem cells. The data, retrieved from the Mouse Cell Atlas, consists of 12009 data points and each sample has 10 feature dimensions, reduced from an original dimension of $23,433$ using Principal Component Analysis (PCA). The sizes of the four clusters[3] are $6506, 2328, 2201, 974$.

**Postures** [$n = 74975, d = 15, K = 5$] (Gardner et al., 2014b;a) comprises images obtained via a motion capture system and a glove for 12 different users performing five hand postures – fist, pointing with one finger, pointing with two fingers, stop (hand flat), and grab (fingers curled). For establishing a rotation and translation invariant local coordinate system, a rigid unlabeled pattern on the back of the glove was utilized. There are a total of 74975 samples in the dataset and the data dimension is 15. The sizes of the given clusters are $19772, 17340, 15141, 12225, 10497$.

**Covtype** [$n = 15120, d = 52, K = 7$] (Blackard & Dean, 1999) comprises digital spatial data for seven forest cover types obtained from the US Forest Service (USFS) and the US Geological Survey (USGS). There are 52 cartographic variables including slope, elevation, and aspect. The dataset has 15120 samples. The sizes of the seven clusters are $3742, 3105, 2873, 2307, 1482, 886, 725$.

**Gaussian** [$n = 30000, d = 10, K = 10$] comprises ten clusters, each generated from a 10-variate Gaussian distribution centered at uniformly at random chosen locations in the unit hypercube. From each cluster, 3000 samples are taken, for a total of 30000 samples. Each Gaussian cluster is spherical with variance 0.5.

**FEMNIST** [$n = 36725, d = 784, K = 62$] (Caldas et al., 2018) is a popular FL benchmark dataset comprising images of digits (0-9) and letters from the English alphabet (both upper and lower cases) from over 3500 users. It dataset is essentially built from the Extended MNIST repository (Cohen et al., 2017) by partitioning it on the basis of the writer of the digit/character. We extract data corresponding to 100 different clients, each of which contributed at least 350 data points. Each image has dimension 784. The size of the largest cluster is 1234, and that of the smallest cluster is 282.

**TCGA** [$n = 1904, d = 57, K = 4$] methylation consists of methylation microarray data for 1904 samples from The Cancer Genome Atlas (TCGA) (Hutter & Zenklusen, 2018) corresponding to four

---

[3]The clusters are obtained by running centralized $K$-means++ clustering multiple times and selecting the one inducing the lowest objective value.

different cancer types: Low Grade Glioma (LGG), Lung Adenocarcinoma (LUAD), Lung Squamous Cell Carcinoma (LUSC) and Stomach Adenocarcinoma (STAD). The observed features correspond to a subset of $\beta$ values, representing the coverage of the methylated sites, at $57$ locations on the promoters of $11$ different genes (ATM, BRCA1, CASP8, CDH1, IGF2, KRAS, MGMT, MLH1, PTEN, SFRP5 and TP53). This subset of genes was chosen for its relevance in carcinogenesis. The sizes of the four clusters are $735, 503, 390, 276$.

**TMI** $[n = 1126, d = 984, K = 4]$ contains samples from human gut microbiomes. We retrieved 1126 human gut microbiome samples from the NIH Human Gut Microbiome (Peterson et al., 2009). Each data point is of dimension 983, capturing the frequency (concentration) of identified bacterial species or genera in the sample. The dataset can be roughly divided into four classes based on gender and age. The sizes of the four clusters are $934, 125, 46, 21$.

## M.2 BASELINE SETUPS.

We use the publicly available implementation of K-FED and DC-KM as our baseline methods. For DC-KM, we set the height of the computation tree to 2, and observe that the leaves represent the clients. Since K-FED does not originally support data removal, has high computational complexity, and its clustering performance is not comparable with that of DC-KM (see Tab. 1), we thus only compare the unlearning performance of MUFC with DC-KM. During training, the clustering parameter $K$ is set to be the same in both clients and server for all methods, no matter how the data was distributed across the clients. Experiments on all datasets except FEMNIST were repeated 5 times to obtain the mean and standard deviations, and experiments on FEMNIST were repeated 3 times due to the high complexity of training. Note that we used the same number of repeated experiments as in Ginart et al. (2019).

## M.3 ENABLING COMPLETE CLIENT TRAINING FOR MUFC

Note that both K-FED and DC-KM allow clients to perform full $K$-means++ clustering to improve the clustering performance at the server. Thus it is reasonable to enable complete client training for MUFC as well to compare the clustering performance on the full datasets. Although in this case we need to retrain affected clients and the server for MUFC upon each removal request, leading to a similar unlearning complexity as DC-KM, the clustering performance of MUFC is consistently better than that of the other two baseline approaches (see Tab. 2). This is due to the fact that we utilize information about the aggregated weights of client centroids.

Table 2: Clustering performance of different FC algorithms compared to centralized $K$-means++ clustering.

|  |  | TMI | Celltype | Gaussian | TCGA | Postures | FEMNIST | Covtype |
|---|---|---|---|---|---|---|---|---|
| | MUFC | **1.05 ± 0.01** | **1.03 ± 0.00** | **1.02 ± 0.00** | **1.02 ± 0.01** | **1.02 ± 0.00** | **1.12 ± 0.00** | **1.02 ± 0.00** |
| Loss ratio | K-FED | 1.84 ± 0.07 | 1.72 ± 0.24 | 1.25 ± 0.01 | 1.56 ± 0.11 | 1.13 ± 0.01 | 1.21 ± 0.00 | 1.60 ± 0.01 |
| | DC-KM | 1.54 ± 0.13 | 1.46 ± 0.01 | **1.02 ± 0.00** | 1.15 ± 0.02 | 1.03 ± 0.00 | 1.18 ± 0.00 | 1.03 ± 0.02 |

## M.4 LOSS RATIO AND UNLEARNING EFFICIENCY

In Fig. 4 we plot results pertaining to the change of loss ratio after each removal request and the accumulated removal time when the removal requests are adversarial. The conclusion is consistent with the results in Section 6.

## M.5 BATCH REMOVAL

In Fig. 5 we plot the results pertaining to removing multiple points within one removal request (batch removal). Since in this case the affected client is more likely to rerun the $K$-means++ initialization for each request, it is expected that the performance (i.e., accumulated removal time) of our algorithm would behave more similar to retraining when we remove more points within one removal request, compared to the case in Fig. 3 where we only remove one point within one removal request.

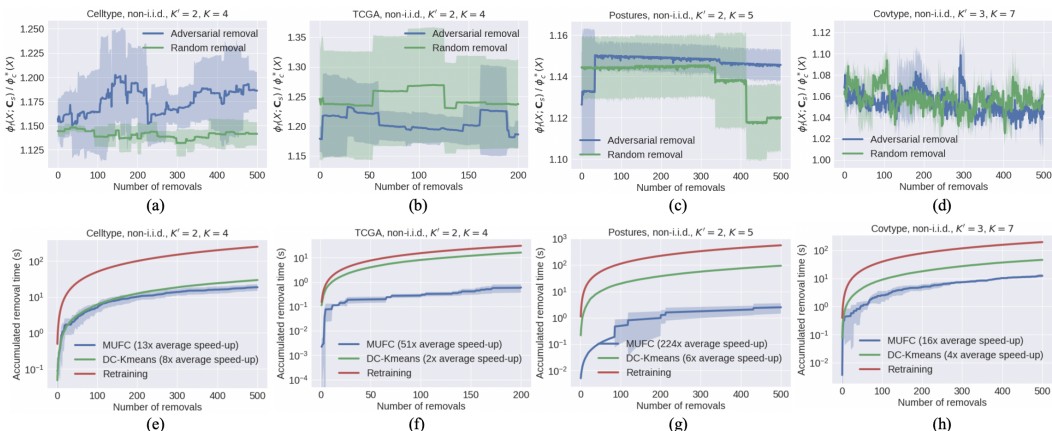

Figure 4: The shaded areas represent the standard deviation of results from different trails for all subplots. (a)-(d) The change of loss ratio $\phi_f(\mathcal{X}; \mathbf{C}_s)/\phi_c^*(X)$ after each round of unlearning procedure. (e)-(h) The accumulated removal time for adversarial removals.

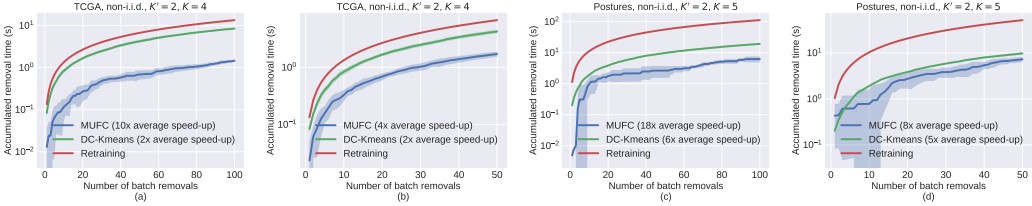

Figure 5: The shaded areas represent the standard deviation of results from different trails for all subplots. (a), (c) Remove 10 points within one batch removal request. (b), (d) Remove 30 points within one batch removal request.

