# OpenReview forum: "Machine Unlearning of Federated Clusters"
_ICLR.cc/2023/Conference — ICLR 2023 poster_

### Official Review · Reviewer_ci9j · 2022-10-21

**Confidence:** 1
**Correctness:** 4
**Technical Novelty And Significance:** 3
**Empirical Novelty And Significance:** 3
**Recommendation:** 6

**Clarity, Quality, Novelty And Reproducibility:**

The experiments are in high quality. For example, authors used shaded areas represent the standard deviation so it is very clear how significant is the improvement.

Reproducibility: authors provided detailed instruction for the script as well as the source code, so I believe it is reproducible. (Though I didn't fully reproduce it by myself).

**Details Of Ethics Concerns:**

In Section 7 ETHICS STATEMENT authors declared they adhered to all regulations, so I believe there is no ethic concerns.

**Strength And Weaknesses:**

My major concern is the benchmark comparison. Authors claim it is the first unlearning algorithm for federate cluster, so it makes hard to understand how innovative compared with peer works

**Summary Of The Paper:**

This work proposes the first known unlearning mechanism for federated clustering with privacy criteria that support simple, provable, and efficient data removal at the client and server level, by combining special initialization procedures with quantization methods that allow for secure aggregation of estimated local cluster counts at the server unit.

In the empirical studies, results show it can achieve 84x across seven datasets compared to completely retraining K-means++ locally and globally for each removal request

**Summary Of The Review:**

Overall I believe it is an interesting paper, with clear writing as well as high quality experiment. My major concern is the benchmark comparison. Authors claim it is the first unlearning algorithm for federate cluster, so it makes hard to understand how innovative compared with peer works. So I choose marginal above threshold with low confidence.

---

> ### Author Response · Authors · 2022-11-12
> **Author Response**
>
> ### "My major concern is the benchmark comparison. Authors claim it is the first unlearning algorithm for federate cluster, so it makes hard to understand how innovative compared with peer works."
>
> We thank the Reviewer for their time and valuable comments.
>
> As summarized in our general response, the contributions of our paper are four-fold and we would appreciate it if the Reviewer could go over them again to see that our manuscript describes novel problems and solutions for the same. With regard to benchmarking, it is always the case that when a new problem is proposed and solved, no other methods can be used for comparison. Still, we try to compare our scheme, which at its core has a federated clustering pipeline, with other federated clustering algorithms such as K-FED [1]. These methods usually do not address at least one of the FL constraints (privacy issues are often pushed aside), so even this comparison may not be fair to our method. Despite all these factors, our method performs exceptionally well in terms of clustering accuracy, communication cost (and, clearly, privacy). For benchmarking, we use data sets used by researchers who have previously reported unlearning methods (such as [2]), but also newly curated data sets from repositories such as TCGA (The Cancer Genome Atlas), which are highly sensitive and significantly more likely to be requested for unlearning than any other data set. Ideally, we would have liked to include more massive data sets in our analysis, but many large-scale biological data sets require special permissions to be used and reported (as one can expect, due to privacy issues). We plan to obtain these for future follow-up work and believe that even at this level, given that we have both the results of theoretical analyses and empirical studies on several data sets, we have demonstrated the excellent potential of our scheme for cluster unlearning tasks.
>
> We hope that the above explanations address your concerns and will be happy to answer any follow-up questions.
>
> [1] Don Kurian Dennis, Tian Li, and Virginia Smith. Heterogeneity for the win: One-shot federated clustering. In International Conference on Machine Learning, pp. 2611–2620. PMLR, 2021.
>
> [2] Antonio Ginart, Melody Guan, Gregory Valiant, and James Y Zou. Making AI forget you: Data deletion in machine learning. Advances in neural information processing systems, 32, 2019.

---

### Official Review · Reviewer_BrQd · 2022-10-22

**Confidence:** 4
**Correctness:** 2
**Technical Novelty And Significance:** 4
**Empirical Novelty And Significance:** 2
**Recommendation:** 6

**Clarity, Quality, Novelty And Reproducibility:**

The reviewer thinks the topic in this paper is very interesting and is of importance in practice. The proposed method is novel to the best of the reviewer's knowledge. However, the description of the method is not clear, the reviewer found it not easy to follow the logic flow of the paper.

**Strength And Weaknesses:**

The paper discusses a new topic of federated unlearning for the purpose of clustering. Given the rich content of the paper, the reviewer believe additional work needs to be done to make the story easier to follow. The reviewer has several suggestions:

1. There are several places in the paper that a term is described first and the definition is given several sections later in the paper, which brings difficulty for the reader to follow.
2. The reviewer thinks it would be useful to first introduce the intuitions behind the algorithms and then present the algorithm. This is particularly the case for algorithm 2.
3. In section 4.2, the reviewer feel a little bit lost since the outliers are introduced in the paper. The discussion of outlier removal is not very closely related to the existence of outliers. Given the space limit of the paper, it would be clearer to focus only on one topic and brings other issues in the appendix or the discussion section.

Major concerns:
1. The reviewer has several questions in terms of algorithm 2 that needs further clarification.
- The quantization of $C^{(l)}$ is performed dimension-wise, can the authors be more specific on how step 4 is performed?
- There is a well known label switching issue under clustering, in step 5 to sum up $q^{(l)}$ at the server, will there be any issue if the label is switched? The reviewer believes switching label at the first place will lead to different $q^{(l)}$ at the first place and hence $q$, this could potentially be a severe issue for clustering.

2. For removing adversarial points, will the cost be necessarily smaller than refitting the model? This is not discussed as the magnitude of $\epsilon_2$ and $s_{\text{min}}$ is not specified in the main paper.

Minor issues:

1. Could the author ellaborate more on the difference between $\phi_f$ and $\phi_{c}$? The value of these two terms are equal when $C=C_s$, are there any particular reason to define two terms?

2. Can the authors ellaborate more on the role of the hyperparameters $\alpha$ and $p$ in the proposed method?

**Summary Of The Paper:**

This paper proposes a novel approach for unlearning data under federated learning setting for the purpopse of clustering. The proposed method also has security gaurantees. Theoretical analysis in terms of clustering performance and complexity analysis has been done for the proposed method.

**Summary Of The Review:**

The reviewer thinks the topic in this paper is very interesting and is of importance in practice. However, there are several concerns that is critical to the proposed method. Therefore, the reviewer believe additional work or clarification is needed before the work can be published.

---

> ### Author Response · Authors · 2022-11-12
> **Author Response**
>
> We thank reviewer BrQd for their review of our manuscript and for providing valuable suggestions. We are currently updating our manuscript so as to make the revised paper easier to follow and understand, and we will be uploading the revision within a few days.  These changes will be marked as blue in the revision.
>
> ### "There are several places in the paper that a term is described first and the definition is given several sections later in the paper, which brings difficulty for the reader to follow."
>
> We apologize for this oversight and will carefully go through the manuscript to find and correct such issues.
>
> ### "The reviewer thinks it would be useful to first introduce the intuitions behind the algorithms and then present the algorithm. This is particularly the case for algorithm 2."
>
> We agree with the reviewer and would have done the same thing if space would have permitted. We are in the process of moving some parts of the main text to free up space for previews of results/algorithms.
>
> ### "In section 4.2, the reviewer feel a little bit lost since the outliers are introduced in the paper. The discussion of outlier removal is not very closely related to the existence of outliers. Given the space limit of the paper, it would be clearer to focus only on one topic and brings other issues in the appendix or the discussion section."
>
> We thank the reviewer for pointing out this issue. We agree that our discussion about outliers could have been more extensive, and the revision will contain more relevant explanations in the main text. Due to space limitations, we could not include the relevant results from Appendix D in the main text, but that is where we made use of Assumption 4.3 about outliers in our proofs. Here we would like to make some extra clarifications about this part.
>
> In a nutshell, we want to address the issue of removal requests that would force frequent retraining from scratch. Given that K-means++ chooses centroids based on their distances from previously selected centroids, points at a fairly large distance to the main cluster structures are frequently selected as centroids. Since outliers by definition are at large distances from other points, they are very likely to lie in the initial set of centroids. Therefore, requesting the removal of an outlier would very likely trigger retraining. Hence, to determine the probability of retraining we need to know how common outliers are in the data set. This means that the notion of outliers is used only for the theoretical analysis of the retraining frequency. Furthermore, outliers can be seen as “adversarial” selections for removal. Perhaps the choice of the word “adversarial” may not have been optimal (as it does not indicate malicious intent but instead suggests that it will have an adversarial effect on retraining), but we hope that the explanation is now clarified.
>
> In addition, the above explanation justifies using the condition “the probability of the data set containing at least one outlier is upper bounded by $O(1/n)$” in the statement of our Lemma 4.4 and Theorem 4.5, and the discussion about outlier removal was originally included to assert that this condition is not overly restrictive.
>
> We sincerely hope that the above explanations clarify the reason behind the introduction of outliers. We are happy to answer any follow-up questions.
>
> ### "The reviewer has several questions in terms of algorithm 2 that needs further clarification."
>
> We will add some intuitions about Algorithm 2 in our revision. We would also like to point the reviewer to Figure 1 for an overview of our Algorithm 2.

---

> > ### Author Response · Authors · 2022-11-12
> > **Author Response**
> >
> > ### "The quantization of $\mathbf{C}^{(l)}$ is performed dimension-wise, can the authors be more specific on how step 4 is performed?"
> >
> > We thank the reviewer for the comment. For simplicity, we assume that the data points lie in a unit square in a two-dimensional Euclidean space (see Figure 1 for the visualization). First, all clients use the same uniform quantization scheme, leading to a set of quantization bins (or grid bins). The quantization procedure will map the centroids in $\mathbf{C}^{(l)}$ to the closest center of these quantization bins. Since the quantization step size is known to both the clients and the server, we can use the index of the (quantization) bins instead of the spatial location of the bin centers to describe the bins. For example, the quantization bin on the top left corner is represented by (row=1, column=1). In a nutshell, we are essentially mapping the bins to a matrix (initialized with 0s), as shown in Figure 1. Next, we record the client's local cluster size in the corresponding bin of this quantization matrix. For example, in Figure 1, we record the value 5 in the entry indexed by (1,1) for client 1, since the green cluster has 5 points. This procedure is performed for all local clusters and on all client data. Thus, each client is in the possession of a sparse matrix representing the bin occupancies, and $q_j^{(l)}$ is just the flattened matrix of client $l$ (i.e., row concatenation). This is the idea behind Step 4 of Algorithm 2. Note that this procedure extends to the case of $d$-dimensional spaces as well.
> >
> > ### "There is a well known label switching issue under clustering, in step 5 to sum up $q^{(l)}$ at the server, will there be any issue if the label is switched? The reviewer believes switching label at the first place will lead to different $q^{(l)}$ at the first place and hence $q$, this could potentially be a severe issue for clustering."
> >
> > Since there is no formal definition of “label switching” in the context of K-means clustering, we assume that the reviewer is either referring to the fact that the labels of data points can change every time we rerun the K-means clustering algorithm, or the label switching problem that could arise when taking a Bayesian approach to parameter estimation and clustering using mixture models [7].
> >
> > For the first case, we would like to point out that this is not an issue for our framework. If the labels of data points on client $l$ change because of retraining, either due to unlearning or system restart, the client information vector $q^{(l)}$ may indeed be different from its previous version. However, this will not affect either the theoretical analyses or the practical performance of Algorithm 2, as different $q^{(l)}$ are just different realizations of multiple runs of the probabilistic K-means++ initialization procedure on client $l$.
> >
> > For the second case, although label switching could be a severe issue when we want to estimate the posterior distribution of the parameters in mixture models (clusters), we believe such an issue never arises in the context of K-means clustering. Since K-means++ is a randomized initialization procedure where the sampling probability only depends on the distances between data points, we do not rely on the knowledge of the posterior distributions of the parameters. Furthermore, in our Algorithm 2, each client performs K-means++ initialization and then uniform quantization. Both procedures only use the information provided by the data points themselves.
> >
> > Please let us know if there is any misunderstanding on our part regarding the question raised, as other potential interpretations of “label switching” are possible.
> >
> > [7] Stephens, Matthew. Dealing with label switching in mixture models. Journal of the Royal Statistical Society: Series B (Statistical Methodology) 62.4 (2000): 795-809.
> >
> > ### "For removing adversarial points, will the cost be necessarily smaller than refitting the model? This is not discussed as the magnitude of $\epsilon_2$ and $s_{\text{min}}$ is not specified in the main paper."
> >
> > We thank the reviewer for this question. Our constants $\epsilon_1$ and $\epsilon_2$ are assumed to be much smaller than the size of the data set $n$. According to our Algorithm 1, the worst-case scenario for adversarial removal is still retraining. So if we remove $R$ points within one removal request, the actual computational cost for random removals is $\text{min}\\\{O(RK^2d), O(nKd)\\\}$, and the cost for adversarial removals is $\text{min}\\\{O(RK^3\epsilon_1\epsilon_2 d), O(nKd)\\\}$. So, the costs of both types of removals are always upper bounded by the costs of retraining. We omitted these details due to space limitations, but we will clarify this part of the text in our revision.

---

> > > ### Author Response · Authors · 2022-11-12
> > > **Author Response**
> > >
> > > ### "Could the author elaborate more on the difference between $\phi_f$ and $\phi_c$? The value of these two terms are equal when $\mathbf{C}=\mathbf{C}_s$, are there any particular reason to define two terms?"
> > >
> > > We thank the reviewer for this observation. We would like to point out that $\phi_f$ and $\phi_c$ are not equal even if the centroid sets are the same, i.e., if  $\mathbf{C}=\mathbf{C}_s$. By definition, we have the centralized objective as $\phi_c(X; C)=\\|X-C\\|_F^2$
> > >
> > > and federated objective as $\phi_f(X; C_s)=\sum_{l=1}^L \\|X^{(l)}-C_s^{(l)}\\|_F^2$ ,
> > >
> > > where $C_s=\begin{bmatrix} C_s^{(1)}\\\\
> > > \cdots \\\\
> > > C_s^{(L)}
> > > \end{bmatrix}.
> > > $
> > >
> > > Note that the definition of the assignment matrix $C\in\mathbb{R}^{n\times d}$ for the centralized K-means is different from that obtained through federated K-means $C_s$: the $i$-th row of $C$ only depends on the location of $x_i$ (the $i$-th data point) while the $i$-th row of $C_s$ depends on the **induced global clustering** described in Definition 5.2.
> > >
> > > One example that explains the difference between these two assignment matrices is as follows. Suppose the global datasets and centroid sets are the same for centralized and federated clustering cases, meaning that we have
> > > $$
> > > X=\begin{bmatrix} X^{(1)}\\\\
> > > \cdots \\\\
> > > X^{(L)}
> > > \end{bmatrix},\\; \mathbf{C}=\mathbf{C}_s=\\{c_1,\dots,c_K\\}.
> > > $$
> > > Suppose that for $x_1$ we have
> > > $$
> > > d(x_1,c_1) < d(x_1,c_j), \forall j\in[K].
> > > $$
> > > Then the first row of $C$ is $c_1$. However, if $x_1$ resides on the memory of client $l$ and belongs to the local cluster $\mathcal{C}_i^{(l)}$, and the recorded local centroid $c_i^{(l)}$ satisfies
> > > $$
> > > d\left(c_i^{(l)},c_2\right) < d\left(c_i^{(l)},c_j\right), \forall j\in[K],
> > > $$
> > > then the first row of $C_s$ is $c_2$, even with $d(x_1,c_1) < d(x_1,c_2)$. This example shows that the assignment matrices $C$ and $C_s$ are different, which also implies that $\phi_f$ and $\phi_c$ are different.
> > >
> > > It is worth pointing out that the notion of an **induced global clustering** is also used in other works such as [1], and the main reason why we adopt this notion when defining our federated objective $\phi_f$ is to be consistent with the existing literature on federated clustering.
> > >
> > > [1] Don Kurian Dennis, Tian Li, and Virginia Smith. Heterogeneity for the win: One-shot federated clustering. In International Conference on Machine Learning, pp. 2611–2620. PMLR, 2021.
> > >
> > > ### "Can the authors elaborate more on the role of the hyperparameters $\alpha$ and $p$ in the proposed method?"
> > >
> > > First, in the distributed K-means clustering literature [8, 9], it is a common strategy to perform oversampling of each sub-dataset to improve clustering performance. Following this idea, we also introduce the oversampling coefficient $\alpha$ in Algorithm 2 to allow for oversampling at each client (i.e., each client samples more than $K$ centroids). As stated in our manuscript, we always assume $\alpha=1$ for simplicity, but in practice one can expect that using larger $\alpha$ will lead to better clustering performance (i.e., to a smaller objective value).
> > >
> > > Second, $p$ is the finite field size with $p\geq \text{max}\\{n, B^d\\}$, where $n$ is the total number of data points, $B=1/\gamma$ is the number of quantization bins for each dimension, and $d$ is the feature dimension. All operations in our SCMA scheme are performed modulo $p$ to make sure that the results lie in the finite field $\mathbb{F}_p$ . See Figure 2 for an example.
> > >
> > > [8] Sudipto Guha, Adam Meyerson, Nina Mishra, Rajeev Motwani, and Liadan O’Callaghan. Clustering data streams: Theory and practice. IEEE transactions on knowledge and data engineering, 15(3): 515–528, 2003.
> > >
> > > [9] Nir Ailon, Ragesh Jaiswal, and Claire Monteleoni. Streaming k-means approximation. Advances in neural information processing systems, 22, 2009.

---

> > > > ### Comment · Reviewer_BrQd · 2022-11-29
> > > > **post-rebuttal**
> > > >
> > > > Thanks for your response. I am happy with most of your responses, just two minor comments.
> > > >
> > > > 1. For the difference between $\phi_f$ and $\phi_c$, I totally understand that the matrices of the centroids are different, my previous comment meant to say that $\phi_f$ can be written as a summation of $\phi_c$s. Introduce new notations for the same thing might lead to some confusion during presentaton. I did not doubt the technical correctness of your notation, but thanks for the clarification.
> > > >
> > > > 2. I still have question with the label switching issue. For label switching, I mean the vectors $q^{(l)}$ across different machines may not be aligned properly.  As you mentioned, "different  are just different realizations of multiple runs of the probabilistic K-means++ initialization procedure on client". Since you need to aggregate the ordered vectors $\{q^{(l)}:l\in [L]\}$ in line 5, what if the order of clusters are different across clients, would this aggregation step have an issue? Could you please explain?

---

> > > > > ### Author Response · Authors · 2022-11-29
> > > > > **Response to post-rebuttal**
> > > > >
> > > > > ### “For the difference between $\phi_f$ and $\phi_c$, I totally understand that the matrices of the centroids are different, my previous comment meant to say that $\phi_f$ can be written as a summation of $\phi_c$s. Introduce new notations for the same thing might lead to some confusion during presentaton. I did not doubt the technical correctness of your notation, but thanks for the clarification.”
> > > > >
> > > > > We thank the reviewer for their suggestion. Unfortunately, given that the period for uploading the revised manuscript with new changes has expired, any notational changes will have to wait until later.
> > > > >
> > > > > ### “I still have question with the label switching issue. For label switching, I mean the vectors $q^{(l)}$ across different machines may not be aligned properly. As you mentioned, "different are just different realizations of multiple runs of the probabilistic K-means++ initialization procedure on client". Since you need to aggregate the ordered vectors ${q^{(l)}:l\in [L]}$ in line 5, what if the order of clusters are different across clients, would this aggregation step have an issue? Could you please explain?”
> > > > >
> > > > > We sincerely thank the reviewer for this insightful question. We now understand what the reviewer was referring to in the context of the label switching problem. We apologize for not understanding it the first time around.
> > > > >
> > > > > A simple answer to the question is that there are no label switching issue of the form suggested by the reviewer. This is due to the fact that we are encoding and transmitting the number of points in local clusters, but not before mapping the centroids of the local clusters to their closest centers of the uniform quantization bin grid.
> > > > >
> > > > > It is possible to envision a distribution of the points from one global cluster across two (or more) clients so that the clients have local clusters whose centroids map to different quantization bin centers. For example, one local cluster can have a centroid mapped to the center of the left-top bin, and another local cluster that corresponds to the same global cluster can have its centroid mapped to the center of the bin right below it. But this is not a problem as at the server side, we securely aggregate the number of points in each bin, but then also subsequently perform an independent global clustering.
> > > > >
> > > > > In a nutshell, we are not relying on the information about local clusters at the server level, but only on the information pertaining to the number of points in each quantization bin. Since the bins all have different identifiers shared by all clients, no confusion can arise regarding the bin identity. Clearly, once the server recovers the number of points in each bin, and performs their own clustering, new global cluster structures will emerge.
> > > > >
> > > > > Furthermore,  there is no ordering of the the $q$-vectors, nor can there be any confusion of what client 1 calls cluster 1, and what client 2 calls cluster 1 (as an example). Both clients securely transmit the indices of bins where their local clusters reside (in space) and there is hence no notion of cluster labels.

---

> > > > > > ### Comment · Reviewer_BrQd · 2022-11-30
> > > > > > **Clarification needed**
> > > > > >
> > > > > > Thank you for your response. I still don't fully understand why "centroids map to different quantization bin centers" is not an issue on the server side. Could you please explain what do you mean by a "pre-defined distribution" in line 6 of algorithm 1? I'm thinking about the following example, for the two matrices in 4th plot of figure 1, we have
> > > > > > | 5 | 10 | 0 |
> > > > > > |---|---|---|
> > > > > > | 0  | 0 | 0 |
> > > > > > | 0  | 0 | 0 |
> > > > > >
> > > > > > and
> > > > > > | 5 | 0 | 0 |
> > > > > > |---|---|---|
> > > > > > | 0  | 10 | 0 |
> > > > > > | 0  | 0 | 0 |
> > > > > >
> > > > > > These two matrices are possible based on your description. Then the aggregation gives
> > > > > > | 10 | 10 | 0 |
> > > > > > |---|---|---|
> > > > > > | 0  | 10 | 0 |
> > > > > > | 0  | 0 | 0 |
> > > > > >
> > > > > > Could you please explain the clustering step on the server? Including from which "pre-defined distribution" that you generate samples?

---

> > > > > > > ### Author Response · Authors · 2022-11-30
> > > > > > > **Clarification for the example**
> > > > > > >
> > > > > > > We thank the reviewer for this example. Once the server receives the aggregated matrix, it performs the clustering as follows. Assume we are using the data generation scheme described on page 5, then the server will have 3 points, each with weight=10, to continue with the clustering with the number of clusters $K=2$ and finally get 2 global centroids. In this case, the "pre-defined distribution" is to choose the center of the quantization bin as the representative point and assign weight $q_j$ to it.
> > > > > > >
> > > > > > > Besides the scheme stated above, we can also generate data points uniformly at random within the quantization bin at the server (Appendix H). In this case, the server will have 30 points, each with weight=1. And they are all generated uniformly at random within the quantization bin they belong to. Then the server can continue the clustering with $K=2$ over these 30 points and finally get 2 global centroids. The "pre-defined distribution" here is the uniform distribution.
> > > > > > >
> > > > > > > In summary, the clustering at the server side purely depends on the aggregated matrix it gets, and it should not be affected by the local cluster labels of each client.
> > > > > > >
> > > > > > > We hope that the above explanation can address the concerns and we are happy to answer any follow-up questions.

---

> > > > > > > > ### Comment · Reviewer_BrQd · 2022-11-30
> > > > > > > > **Final response**
> > > > > > > >
> > > > > > > > Thanks for the clarification, I have confirmed my understanding is correct. My concern is that in such cases the clusters at the server may not be the same as what we anticipated, but this is a minor issue and I hope this can be addressed in the future version. After the discussion, I tend to accept the paper based on its novelty. I have updated my score accordingly.

---

> > > > > > > > > ### Author Response · Authors · 2022-12-01
> > > > > > > > > **Thank you for your comments**
> > > > > > > > >
> > > > > > > > > We sincerely appreciate your active participation in the discussion and your insightful comments. We will address your concern regarding the minor issue in the revised manuscript.

---

### Official Review · Reviewer_5qA2 · 2022-10-25

**Confidence:** 4
**Clarity, Quality, Novelty And Reproducibility:** It is understandable to a large exten…
**Correctness:** 2
**Technical Novelty And Significance:** 2
**Empirical Novelty And Significance:** 2
**Recommendation:** 3

**Strength And Weaknesses:**

Strength:
1.The paper proposes a unlearning mechanism for federated clustering with privacy criteria that enables  efficient data removal at the client and server level.

2. The extensive theoretical analyses are derived to validate the performance guarantees of the proposed method.

3. The algorithm is evaluated on a number of datasets to demonstrate the effectiveness of the proposed approach for machine unlearning.

Weakness:
1.	According to algorithms 1 and 4, the unlearning process requires the help of removal set XR. This requirement is naturally fulfilled in the context of centralized unlearning but may not be available. In such case, the retraining in line 3 in algorithm 4 will always be required and significantly increase the complexity.
2.	The unlearning strategy in this paper relies heavily on the k-means++ algorithm, which approximates the k-means problem proposed in 2007. The paper exploits the nature of k-means++ initialization with no technical improvement, making the novelty and contribution questionable.
3.	Line 2 in algorithm 1 implies that the centroid set C can remain the same if the removal point is not inside the centroid set C without further restrictions. This is very questionable when a certain number of biased data points are removed within a single request. In the unlearning experiment setting, the unlearning requests are carried out with one data point per client at each round of unlearning, which is insufficient to evaluate the performance of the proposed unlearning method.

**Summary Of The Paper:**

This paper focuses on unlearning federated clustering problems with k-means++ initialization and secure compressed multiset aggregation. The idea of unlearning is to perform k-means++ initialization at each local client. By exploiting the nature of k-means++ approximation, the centroid set C may remain unchanged if removed points are not in C.

**Summary Of The Review:**

In general, the studied problem is interesting and important. In addition, the methodology is principled with three major merits as discussed above. However, the work still has some unaddressed concerns to well justify its technical and empirical contributions.

---

> ### Author Response · Authors · 2022-11-12
> **Author Response**
>
> We thank Reviewer 5qA2 for their valuable review and their positive comments on the work. In what follows, we would like to clarify certain aspects of our algorithms, results, and contributions, and address the concerns raised.
>
> ### "According to algorithms 1 and 4, the unlearning process requires the help of removal set $\mathcal{X}_R$. This requirement is naturally fulfilled in the context of centralized unlearning but may not be available. In such case, the retraining in line 3 in algorithm 4 will always be required and significantly increase the complexity."
>
> We would like to clarify that in our federated clustering setting, training data only resides at the client level (e.g., patient data is only stored at hospitals), so the data removal requests will also only arise on the client side. Our assumption, which is also the assumption in most of the unlearning literature, is that the affected client $l$ will know which points are requested to be removed (the removal set $\mathcal{X}_R^{(l)}$), but that the server and other clients will not require or receive any information about $\mathcal{X}_R^{(l)}$. Based on our federated clustering framework, the server can only know the processed information sent by the clients, and the model updates at the server side do not depend on $\mathcal{X}_R^{(l)}$. Furthermore, as our proposed federated clustering algorithm is a one-shot algorithm, unlearning is only needed for the local client models affected by the requests (centroids $\mathbf{C}^{(l)}$) and the server model (centroids $\mathbf{C}_s$). The models of all other clients that did not receive removal requests will not be affected and thus will not change.
>
> Additionally, our unlearning procedure for federated clustering (Algorithm 4) works as follows: Client $l$ will first run Algorithm 1, meaning that client $l$ will only rerun the initialization step (retrain) when $\mathbf{C}^{(l)}\cap \mathcal{X}_R^{(l)} \neq \varnothing$; If client $l$ updates its local centroids and the corresponding information vector $q^{(l)}$, it will notify the server and the server will repeat the SCMA aggregation procedure with the new counts from affected clients to obtain the new aggregated vector $q^\prime$; once the server generates the new vector $q^\prime$, it will rerun the full K-means++ algorithm to obtain new centroids $\mathbf{C}_s^\prime$. So, in our algorithm (line 3 of Algorithm 4), client $l$ will retrain **only when** $\mathbf{C}^{(l)}\cap \mathcal{X}_R^{(l)} \neq \varnothing$. Note that if retraining is not needed here, the computational complexity for Algorithm 1 is just $O(RK)$, which is significantly smaller than the complexity of retraining, $O(n^{(l)}Kd)$, given that $R \ll n^{(l)}$.
>
> We hope that the above explanations can help clarify our Algorithm 4, and we are happy to answer any follow-up questions the reviewer may have.
>
> ### "The unlearning strategy in this paper relies heavily on the k-means++ algorithm, which approximates the k-means problem proposed in 2007. The paper exploits the nature of k-means++ initialization with no technical improvement, making the novelty and contribution questionable."
>
> All our contributions of this work are summarized in our general response at the beginning, and we would like to elaborate on some of them for clarity. First, we did not try to claim that the use of K-means++ is an important contribution of our work. We are using K-means++ as is since it is a powerful algorithm that is also exceptionally well-suited for federated clustering and unlearning in particular, both from the theoretical and practical perspective. Note that K-means++ is just one tiny component in our end-to-end design of the first known federated unlearning system. What is completely novel in our system is the fact that our work is the first to provide a comprehensive framework that enables efficient unlearning of points for unsupervised clustering problems in the fast-growing privacy-preserving federated learning setting.
>
> --> To be continued on the next block

---

> > ### Author Response · Authors · 2022-11-12
> > **Author Response**
> >
> > Next, we would like to restate our contributions in more detail. One of our major contributions is to provide a comprehensive federated clustering framework with extensive theoretical analyses on both the clustering performance and the computational and communication complexity. We also show that our federated clustering algorithm exhibits superior clustering performance across multiple real-world datasets compared to a handful of previously reported methods, including K-FED [1] (ICML’21) and DC-Kmeans [2] (Neurips’19). Please see Appendix N.3 and Section 6 for detailed comparisons. Within our federated clustering framework, we also develop a novel scheme, secure compressed aggregation scheme (SCMA), which is an aggregation scheme that has low communication complexity but ensures **local data privacy** and **exact aggregated data recovery**. For example, suppose that the local models at the clients are denoted by $q^{(l)}$, and that we need to retrieve $q=\sum_{l=1}^L q^{(l)}$ at the server. Most existing communication-efficient aggregation algorithms require the server to know exact information about each individual $q^{(l)}$ to generate $q$ or its approximated version. On the other hand, our SCMA scheme ensures that the server knows nothing about $q^{(l)}$ beyond what is revealed from the aggregate $q=\sum_{l=1}^L q^{(l)}$. Note that with SCMA, the server always retrieves the exact sum $q$. Any approximations of $q$ at the server may result in inferior final clustering performance and lead to an intractable theoretical analysis. Furthermore, SCMA ensures that the server is able to perform secure aggregation of sparse vectors across clients and obtain an exact sum of the sparse vectors with a communication complexity that is **logarithmic** in the vector dimension, significantly outperforming the prior state-of-the-art methods [3, 4] for sparse secure aggregation with a communication complexity that is **linear** in the vector dimension. Thus, SCMA is an independent and novel contribution by itself to the FL community.
> >
> > Another major contribution of our work to the area of unlearning is a rigorous analysis that establishes the close connection between the exact unlearning and the K-means++ probabilistic initialization procedure, which allows us to develop efficient unlearning methods for clustering models. We also introduce, for the first time, the notion of adversarial removals into the unlearning literature, which is much harder to analyze. Furthermore, we analyze the removal time complexity for both random and adversarial removals and provide theoretical guarantees for the unlearning complexity.
> >
> > Finally, we compiled a collection of datasets for benchmarking unlearning of federated clusters, including two new datasets which contain anonymized cancer methylation patterns and gut microbiome information. We believe that developing the first unlearning benchmarks for these medical datasets, especially in the context of federated clustering, is a contribution of significant interest to computational biologists and medical research community.
> >
> > Given all our contributions highlighted above, we believe that our method should not be viewed as limited in novelty and technical content. We will be happy to answer any follow-up questions.
> >
> > [1] Don Kurian Dennis, Tian Li, and Virginia Smith. Heterogeneity for the win: One-shot federated clustering. In International Conference on Machine Learning, pp. 2611–2620. PMLR, 2021.
> >
> > [2] Antonio Ginart, Melody Guan, Gregory Valiant, and James Y Zou. Making AI forget you: Data deletion in machine learning. Advances in neural information processing systems, 32, 2019.
> >
> > [3] Constance Beguier, Mathieu Andreux, and Eric W Tramel. Efficient sparse secure aggregation for federated learning. arXiv preprint arXiv:2007.14861, 2020.
> >
> > [4] Irem Ergun, Hasin Us Sami, and Basak Guler. Sparsified secure aggregation for privacy-preserving federated learning. arXiv preprint arXiv:2112.12872, 2021.
> >
> > ### "Line 2 in algorithm 1 implies that the centroid set C can remain the same if the removal point is not inside the centroid set C without further restrictions. This is very questionable when a certain number of biased data points are removed within a single request. In the unlearning experiment setting, the unlearning requests are carried out with one data point per client at each round of unlearning, which is insufficient to evaluate the performance of the proposed unlearning method."
> >
> > We would like to provide some clarifications that should hopefully better explain how our algorithm works (we apologize if some important details that clarify the approach were omitted due to space constraints; we will make sure to add clarifications into our revision).
> >
> > --> To be continued on the next block

---

> > > ### Author Response · Authors · 2022-11-12
> > > **Author Response**
> > >
> > > First, as shown in the pseudocode of Algorithm 1, our method supports removing multiple points (denoted as $\mathcal{X}_R$) within a single request. The main idea of Algorithm 1 is that we will only rerun the initialization procedure, which corresponds to retraining, when $\mathbf{C} \cap \mathcal{X}_R \neq \varnothing$; otherwise the centroid set $\mathbf{C}$ remains the same. In a nutshell, if we receive a certain number of removal requests and it turns out that at least one of the requested points lies in $\mathbf{C}$, we immediately retrain. Since the centroids selected by the K-means++ procedure are always true data points, we are guaranteed that the updated $\mathbf{C}^\prime$ will not contain any of the points in $\mathcal{X}_R$ requested for removal. We further show that this unlearning approach satisfies the exact unlearning criterion in Appendix C (Proof of Lemma 4.1), and the conclusion holds for arbitrary removal sets $\mathcal{X}_R$, no matter if they contain biased data points or not.
> > >
> > > Next, for the experiment part, removing one data point at each round of unlearning can in fact best explain the benefits of unlearning and it corresponds to the key motivation of machine unlearning. Note that machine unlearning aims to reduce the frequency of retraining, **not** to completely avoid retraining. In other words, machine unlearning focuses on cases where always retraining is not necessary and wasteful, and removing only one data point at each round is the best example of such cases. However, if a large amount of data points is removed within a single request (the extreme case is to remove 99% of the training data), then it should be optimal for all unlearning methods to perform retraining. This observation also holds for another related but different subject, differential privacy (DP). It is also worth pointing out that removing one data point at each round is a common practice for simulations in both unlearning and DP literature [2, 5, 6]. Nevertheless, we have included the theoretical analysis on removal time complexity for multiple points removal within one removal request in Theorem 4.5, and in practice, we would expect the performance of our unlearning method to behave more similarly to retraining when we remove more points within one removal request.
> > >
> > > We hope our clarifications would answer your comment. Please feel free to raise any follow-up questions if needed.
> > >
> > > [2] Antonio Ginart, Melody Guan, Gregory Valiant, and James Y Zou. Making AI forget you: Data deletion in machine learning. Advances in neural information processing systems, 32, 2019.
> > >
> > > [5] Chuan Guo, Tom Goldstein, Awni Hannun, and Laurens Van Der Maaten. Certified data removal from machine learning models. Proceedings of the 37th International Conference on Machine Learning, PMLR, 2020.
> > >
> > > [6] Dwork, Cynthia. Differential privacy. Encyclopedia of cryptography and security. 2011.

---

### Official Review · Reviewer_j6JM · 2022-10-29

**Confidence:** 3
**Correctness:** 4
**Technical Novelty And Significance:** 3
**Empirical Novelty And Significance:** 3
**Recommendation:** 8

**Clarity, Quality, Novelty And Reproducibility:**

- Clarity: the presentation of the paper is clear, and the proposed method is easy to follow.
- Quality: the paper is well written. Both theoretical and empirical results seem to come with good quality.
- Novelty: the proposed method is novel.
- Reproducibility: enough implementation details are provided to demonstrate the effectiveness of the proposed method.

**Strength And Weaknesses:**

Strength:
- The paper is well-written.
- Solving the unlearning problem for federated tasks is of great importance.
- Both theoretical analysis and experimental results are shown to justify the effectiveness of the proposed method.

Weaknesses:
- It seems the scope of the proposed method is a bit limited as it seems to only be designed for federated clustering. I wonder how hard it will be to generalize the proposed method to general federated unlearning.
- the scale of the experiment is a bit small. What will the performance look like on datasets with the ImageNet scale?
- I am almost sure that SCMA (or at least its variants) has been studied in communication-efficient distributed optimization, e.g., via gradient or model quantization and/or sparsification. This limits the novelty of the proposed method.

**Summary Of The Paper:**

The paper proposes a new federated unlearning method for a specific federated clustering workflow. A key to the federated unlearning method is secure compressed multiset aggregation (SCMA) to aggregate local contributing data and enhance data privacy. Theoretical analysis is shown to demonstrate the computation and communication effectiveness of the proposed method. Experimental results are also shown to justify the practical speedup of the proposed method.

**Summary Of The Review:**

The paper proposes a good and efficient method for federated unlearning of federated clustering tasks. Both theoretical and empirical results are shown to justify its performance. The scope of the proposed method is limited by it only tackles federated clustering problems.

---

> ### Author Response · Authors · 2022-11-12
> **Author Response**
>
> We thank the reviewer j6JM for their time and valuable comments. Our response is provided below.
>
> ### “It seems the scope of the proposed method is a bit limited as it seems to only be designed for federated clustering. I wonder how hard it will be to generalize the proposed method to general federated unlearning.”
>
> Machine unlearning is task-specific, meaning that the unlearning mechanisms strongly depend on the model training procedure. Hence, different problem formulations, model architectures, and training procedures naturally lead to different unlearning methods. Therefore, it is very hard, if at all possible, to design a generalized unlearning framework for **all** types of FL problems. Nevertheless, our work aims to provide a “general” unlearning method for an important problem domain in FL: unsupervised federated clustering. As a matter of fact, to the best of our knowledge, our work is the first one to provide a comprehensive framework with theoretical analyses that enables efficient unlearning of points for federated clustering models. Specifically, we first present a novel federated clustering algorithm that exhibits superior clustering performance across multiple real-world datasets compared to that of the handful of previously reported methods, including K-FED [1] (ICML’21) and DC-Kmeans [2] (Neurips’19). Please see Appendix N.3 and Section 6 for detailed comparisons. Furthermore, we propose an intuitive and efficient unlearning mechanism for the clustering procedure, by exploiting the connection between the exact unlearning and the K-means++ probabilistic initialization procedures. We also illustrate the superior unlearning performance of our method compared to DC-Kmeans, which is the only existing work that can be applied to this problem, again via extensive simulation results.
>
> We sincerely hope that the reviewer will reconsider the score in light of the aforementioned explanations as well as the four contributions of our work summarized in our general response. We will be happy to address any additional questions/concerns as well.
>
> [1] Don Kurian Dennis, Tian Li, and Virginia Smith. Heterogeneity for the win: One-shot federated clustering. In International Conference on Machine Learning, pp. 2611–2620. PMLR, 2021.
>
> [2] Antonio Ginart, Melody Guan, Gregory Valiant, and James Y Zou. Making AI forget you: Data deletion in machine learning. Advances in neural information processing systems, 32, 2019.
>
> ### "The scale of the experiment is a bit small. What will the performance look like on datasets with the ImageNet scale?"
>
> We thank the reviewer for the question. While in principle we agree that larger data sets should be considered, the experiments we present in the manuscript either use data sets previously reported in the unlearning literature (but outside the scope of FL, see [2]) or new data sets from the area of computational biology (since unlearning most frequently happens in this application domain). Also, for large-scale datasets with complex structures like ImageNet, simple clustering-based algorithms are known to result in inferior performance compared to deep neural network methods. An interesting future direction could be to design specialized clustering algorithms for large-scale datasets such as ImageNet or the UK Biobank, which is of similar size as ImageNet but much more likely to receive frequent unlearning requests. Furthermore, as pointed out in our fourth contribution, we also construct two **new** unlearning benchmarks concerning biological data, whose sensitive nature makes them amenable to unlearning requests from users (one contains sensitive cancer methylation markers, retrieved from TCGA which generally does not have data at the scale of ImageNet). We believe that providing the first federated clustering and unlearning benchmarks for multiple real-world data sets in general and two new biological data sets in particular, is quite valuable.

---

> > ### Author Response · Authors · 2022-11-12
> > **Author Response**
> >
> > ### "I am almost sure that SCMA (or at least its variants) has been studied in communication-efficient distributed optimization, e.g., via gradient or model quantization and/or sparsification. This limits the novelty of the proposed method."
> >
> > We thank the reviewer for pointing out this question. To the best of our knowledge, no other known method parallels the features of SCMA. First, the classical Shamir’s secret-sharing method in cryptography appears superficially similar to our approach as both are related to Reed-Solomon (RS) codes. But that is where the similarity ends. Our evaluation points are the bin indices which are all distinct and can hence be assumed to be different elements of a prime order field. Second, the counts of cluster points in the bins represent the coefficients of the evaluation polynomials, again a unique feature of our scheme. SCMA also includes an inherent compression feature that is not present in Shamir-like schemes. On the other hand, we agree that there exist other methods to aggregate local models (i.e., vectors) in the general distributed learning literature, when **data privacy is not a primary concern**, and only a handful of methods exist for sparse secure aggregation that come with higher communication costs compared to our scheme.
> >
> > The main practical advantage of our SCMA scheme is that we can achieve low communication complexity while also ensuring **local data privacy** and **exact aggregated data recovery**. For example, suppose that the local models at the clients are denoted by $q^{(l)}$, and that we need to retrieve $q=\sum_{l=1}^L q^{(l)}$ at the server. Most existing communication-efficient aggregation algorithms require the server to know the individual vectors $q^{(l)}$ to generate $q$ or an approximation for it. On the other hand, our SCMA scheme ensures that the server knows nothing about $q^{(l)}$ beyond what is revealed from the aggregate $q=\sum_{l=1}^L q^{(l)}$. Note that with SCMA, the server always retrieves the exact sum $q$. Any approximations of $q$ at the server may result in inferior final clustering performance and lead to an intractable theoretical analysis. Furthermore, SCMA ensures that the server is able to perform secure aggregation of sparse vectors across clients and obtain an exact sum of the sparse vectors with a communication complexity that is **logarithmic** in the vector dimension, significantly outperforming the prior state-of-the-art methods [3, 4] for sparse secure aggregation with a communication complexity that is **linear** in the vector dimension.
> >
> > Hence, we still believe that our proposed SCMA scheme is the first known scheme that performs **exact** and **communication-efficient** aggregation while **satisfying the predefined privacy criterion**. We are grateful to the reviewer for this comment and will be adding this discussion to our revision to reaffirm the novelty of the SCMA scheme.
> >
> > [3] Constance Beguier, Mathieu Andreux, and Eric W Tramel. Efficient sparse secure aggregation for federated learning. arXiv preprint arXiv:2007.14861, 2020.
> >
> > [4] Irem Ergun, Hasin Us Sami, and Basak Guler. Sparsified secure aggregation for privacy-preserving federated learning. arXiv preprint arXiv:2112.12872, 2021.

---

> > > ### Comment · Reviewer_j6JM · 2022-12-06
> > > **Thanks for the authors' response**
> > >
> > > After carefully reviewing the authors' responses. My major concerns are addressed. More specifically, the authors have convinced me that (i) a uniform solution for all federated unlearning problems is hard, (ii) they have contributed two new federated datasets in the biological domain; and (iii) the SCMA scheme is novel. I will therefore raise my overall evaluation score.

---

> > > > ### Author Response · Authors · 2022-12-07
> > > > **Thanks for your comments**
> > > >
> > > > Thank you for your positive comments. Please feel free to let us know in case you have any further questions.

---

### Author Response · Authors · 2022-11-12
**General Response**

We thank the area chair and reviewers for their time and comments that improve the exposition and readability of our manuscript. We will upload our revision as soon as possible. Also, the summary of our contributions is listed below -- we hope that this overview will resolve issues that were brought up in the context of our technical innovations. More detailed explanations of the contributions are provided to address the questions raised by the reviewers.

1. We propose a novel federated clustering algorithm that offers order-optimal approximation guarantees for the federated K-means clustering objective, and also outperforms the handful of existing related methods [1, 2], especially for the case that the cluster sizes are highly imbalanced (see Appendix N.3 and Section 6 for more details).

2. We develop a novel sparse compressed multiset aggregation scheme that ensures that our federated clustering algorithm satisfies a new privacy criterion defined in Section 3. This privacy criterion asserts that the server only has access to the aggregated counts of points in individual clusters but does not have any information about the point distributions at individual clients, nor about the actual locations of the points within the clustering space. Our SCMA scheme outperforms existing works specially designed for sparse secure aggregation [3, 4] with respect to communication complexity. SCMA securely recovers the exact sum of the input sparse vectors with a communication complexity that is **logarithmic** in the vector dimension. In contrast, [3, 4] have a communication complexity that is **linear** in the vector dimension.

3. We introduce, for the first time, the problem of machine unlearning of federated clusters. Our approach is based on the K-means++ initialization procedure, which we show is ideally suited for unlearning. To show why K-means++ helps with the unlearning task, we perform a new theoretical analysis that shows that the removal time complexity of our method is significantly lower than that of complete retraining, both under random and a new adversarial removal scenario. Note that this work is also the first one to introduce adversarial removals and model the underlying data points as instances that force frequent retraining of the K-means++ initialization procedure.

4. We curate two new unlearning benchmarking data sets including methylation patterns in cancer genomes and gut microbiome information, which may be of significant importance to computational biologists and medical researchers that are frequently faced with unlearning requests.

[1] Don Kurian Dennis, Tian Li, and Virginia Smith. Heterogeneity for the win: One-shot federated clustering. In International Conference on Machine Learning, pp. 2611–2620. PMLR, 2021.

[2] Antonio Ginart, Melody Guan, Gregory Valiant, and James Y Zou. Making AI forget you: Data deletion in machine learning. Advances in neural information processing systems, 32, 2019.

[3] Constance Beguier, Mathieu Andreux, and Eric W Tramel. Efficient sparse secure aggregation for federated learning. arXiv preprint arXiv:2007.14861, 2020.

[4] Irem Ergun, Hasin Us Sami, and Basak Guler. Sparsified secure aggregation for privacy-preserving federated learning. arXiv preprint arXiv:2112.12872, 2021.

---

### Author Response · Authors · 2022-11-20
**Changes in Revision**

We have made the following changes (marked as blue) in our revision:

1. We rewrite part of the abstract and the contribution part to make our contributions clearer.
2. We add a discussion on page 3 to explain the difference between the centralized objective $\phi_c$ and federated objective $\phi_f$.
3. We change the order of Section 4 and Section 5. In our revision, we first describe our FC framework in Section 4, then describe the unlearning mechanism in Section 5.
4. We add intuitions and step-by-step explanations of Algorithm 1 (secure FC) on pages 4 and 5.
5. We add explanations for adversarial removals on page 7.
6. We add intuitions and step-by-step explanations of Algorithm 4 (unlearning FC) on page 7.
7. We add new simulation results on batch removal in Appendix M.5 (Figure 5).
8. We add more discussions about existing works on sparse secure aggregation in Appendix A.

---

### Decision · Program_Chairs · 2023-01-20

**Decision:**

Accept: poster

**Justification For Why Not Higher Score:**

Reviewer j6JM has correctly pointed out that the impact of this work is limited to unlearning in federated clustering rather than a general-purpose unlearning method.

**Justification For Why Not Lower Score:**

See reason (A) above.

**Metareview: Summary, Strengths And Weaknesses:**

The main contribution of this work lies in efficient unlearning for privacy-preserving federated clustering based on the secure compressed multiset aggregation (SCMA) framework that can achieve low communication complexity while ensuring local data privacy and exact aggregated data recovery.

After reviewing and responding to the authors' rebuttal and an active discussion, (A) most reviewers have agreed that the contributions of this work (especially the SCMA framework and theoretical analysis of K-means++ enabling exact unlearning) are novel and interesting.

Some reviewers have initially raised a number of concerns requiring clarifications. The authors have carefully addressed them in the rebuttal and revised paper to alleviate the reviewers' concerns.

**Note From Pc:**

if the above contains the word "oral" or "spotlight" please see: "oral" presentation means -> notable-top-5% and "spotlight" means -> notable-top-25%. As stated in our emails, we are disassociating presentation type from AC recommendations

**Summary Of Ac-Reviewer Meeting:**

There is sufficient **written** discussion generated on the OpenReview discussion forum to the extent of being able to reach a consensus on the recommendation. Hence, there is no need for a meeting.